**Article** https://doi.org/10.1038/s41467-024-45086-5

# The mechanism of low blue light-induced leaf senescence mediated by GmCRY1s in soybean

Zhuang Li[1,2], Xiangguang Lyu[1,2], Hongyu Li[1,2], Qichao Tu[1], Tao Zhao[1], Jun Liu[1] & Bin Liu ⬡ [1] ✉

Leaf senescence is a crucial trait that has a significant impact on crop quality and yield. Previous studies have demonstrated that light is a key factor in modulating the senescence process. However, the precise mechanism by which plants sense light and control senescence remains largely unknown, particularly in crop species. In this study, we reveal that the reduction in blue light under shading conditions can efficiently induce leaf senescence in soybean. The blue light receptors GmCRY1s rather than GmCRY2s, primarily regulate leaf senescence in response to blue light signals. Our results show that GmCRY1s interact with DELLA proteins under light-activated conditions, stabilizing them and consequently suppressing the transcription of *GmWRKY100* to delay senescence. Conversely, LBL reduces the interaction between GmCRY1s and the DELLA proteins, leading to their degradation and premature senescence of leaves. Our findings suggest a GmCRY1s-GmDELLAs-GmWRKY100 regulatory cascade that is involved in mediating LBL-induced leaf senescence in soybean, providing insight into the mechanism of how light signals regulate leaf senescence. Additionally, we generate *GmWRKY100* knockout soybeans that show delayed leaf senescence and improved yield under natural field conditions, indicating potential applications in enhancing soybean production by manipulating the leaf senescence trait.

Leaf senescence is an age-dependent process that is regulated by both internal (leaf age and phytohormones) and external (light, temperature, and stress) factors[1,2]. During senescence, leaf cells experience extensive alterations in gene expression and metabolism, resulting in the decomposition of macromolecules and the reallocation of essential nutrients to newer tissues or reproductive organs[3,4]. Premature leaf senescence can be induced by unfavorable environmental conditions such as prolonged darkness, nutrient deficiency, drought, shading, pathogen infection, and wounding[1,5,6], and may lead to a reduction in crop yield by affecting photosynthetic efficiency and nutrient remobilization. For this reason, exploring the regulatory mechanisms of leaf senescence is necessary not only to understand a fundamental biological process, but also to create new strategies for improving the agricultural properties of crops.

Light is a crucial environmental factor that affects leaf senescence[7]. Plants are able to sense the light environment using an intricate network of photoreceptors, including phytochromes (PHYs) perceiving far-red and red light, cryptochromes (CRYs), phototropins and ZTL-type receptors perceiving blue and ultraviolet-A light, and UV RESISTANCE LOCUS 8 (UVR8) perceiving ultraviolet-B light[8–12]. When a plant is partially shaded by neighboring vegetation, leaves experience accelerated senescence, known as one of the shade avoidance syndromes (SAS)[13,14]. This is detected as a decrease in red to far-red light ratio (low R:FR)[15], which inactivates the PHYs, causing rapid

[1]State Key Laboratory of Crop Gene Resources and Breeding, Institute of Crop Sciences, Chinese Academy of Agricultural Sciences, Beijing, China. [2]These authors contributed equally: Zhuang Li, Xiangguang Lyu, Hongyu Li. ✉e-mail: liubin05@caas.cn

dephosphorylation and accumulation of phytochrome-interacting factors (PIFs), and ultimately inducing premature leaf senescence[16–18]. In *Arabidopsis thaliana*, the absence of the *PIF* genes significantly delayed leaf senescence, whereas overexpression of *PIF* genes accelerated both age-dependent and dark-induced senescence[19]. PIFs also function in the signaling pathways of the senescence-promoting hormones, including ethylene and abscisic acid, by directly activating the expression of *ETHYLENE-INSENSITIVE 3* (*EIN3*), *ENHANCED EM LEVEL* (*EEL*), and *ABA INSENSITIVE 5* (*ABI5*)[20]. These genes directly activate the expression of the major senescence-promoting NAC transcription factor (TF) *ORESARA1* and chlorophyll degradation regulatory gene *NON-YELLOWING1* (*NYE1*), and repress the chloroplast activity maintainer gene *GOLDEN 2-LIKE2* (*GLK2*) by binding to their promoter regions[21,22].

Phytohormones are the most critical endogenous components known to control the progression of leaf senescence in plants[1,6,23]. Among them, cytokinin and auxin inhibit leaf senescence, while ethylene, salicylic acid (SA), abscisic acid (ABA), and jasmonate (JA) accelerate its aging[24–27]. However, the role of gibberellin (GA) in leaf senescence is uncertain. GA is vital for many developmental processes in plants, such as seed germination, stem elongation, and floral transition[28]. GA signaling is detected and transduced by the GA-GID1 (GIBBERELLIN INSENSITIVE DWARF1)-DELLA regulatory module. DELLA proteins, including GA INSENSITIVE (GAI), REPRESSOR OF ga1-3 (RGA), RGA-LIKE 1 (RGL1), RGL2, and RGL3, are negative regulators of GA signaling and have distinct and overlapping roles in *Arabidopsis*[29–31]. Studies have demonstrated that DELLA protein RGA interacts with WRKY6 and negatively regulates dark-induced leaf senescence and chlorophyll degradation[32,33]. RGL1 also functions as a negative regulator of leaf senescence, repressing the transcriptional activation activity of WRKY45 on several senescence-associated genes (*SAGs*), including *SAG12*, *SAG13*, *SAG113*, and *SEN4*[34]. However, the role of DELLA proteins in regulating leaf senescence in other species and whether DELLA proteins are involved in regulating light-mediated leaf senescence remain unknown.

Studies have established the roles of WRKY transcription factors (TFs) in regulating biotic and abiotic stress responses, as well as multiple developmental and physiological processes[35–37]. In *Arabidopsis*, WRKYs have been identified as important regulators of leaf senescence. For example, WRKY6 is known to positively regulate leaf senescence by specifically activating the expression of *senescence-induced receptor-like kinase* (*SIRK*) gene[38]. Meanwhile, WRKY75 has been shown to accelerate the progression of leaf senescence by promoting the transcription of *SA INDUCTION-DEFICIENT2* (*SID2*) to increase SA content and inhibiting the transcription of *CATALASE2* (*CAT2*) to reduce $H_2O_2$ scavenging[39]. WRKY28 also plays a role in high R:FR-induced leaf senescence. FHY3, the key regulator of the phytochrome A-mediated signaling pathway, directly binds to the promoter region of *WRKY28* to suppress its expression under high R:FR light conditions, thereby negatively regulating SA biosynthesis and leaf senescence[40].

In contrast to the well-documented mechanism of low R:FR-induced leaf senescence in the model plant *Arabidopsis*, the physiological process of LBL-induced leaf senescence remains poorly understood. CRYs, as the primary blue-light receptors play essential roles in photomorphogenesis and photoperiodic flowering[8,41,42], appear to have no apparent influence on leaf senescence in *Arabidopsis*[21]. A study on soybean revealed that GmCRY2a negatively regulates leaf senescence by interacting with the basic helix-loop-helix transcriptional factor GmCIB1 (cryptochrome-interacting bHLH1) and inhibiting its transcriptional activity on *SAGs*[43]. Additionally, LBL induces obvious SAS including exaggerated stem elongation in soybean[44], suggesting that soybean may be a suitable organism to study the mechanisms of LBL-induced leaf senescence in plants.

In this study, we demonstrate that LBL can induce clear leaf senescence in soybean. We find that, under normal light conditions, GmCRY1s interact with and stabilize the DELLA proteins GmRGAa and GmRGAb, which directly inhibit the transcription of the senescence activator *GmWRKY100*. However, LBL disrupts this GmCRY1s-DELLA interaction, leading to the degradation of DELLA proteins and upregulation of *GmWRKY100* transcription, thus promoting leaf senescence. Therefore, we uncover a GmCRY1s-GmDELLAs-GmWRKY100 signaling module to explain the mechanisms by which soybean perceives LBL shade signals and initiates leaf senescence.

## Results

### LBL induces premature leaf senescence in soybean

Dense planting and intercropping cultivations can induce typical symptoms of shade avoidance syndrome (SAS), which includes exaggerated stem elongation and premature leaf senescence in soybean[44–46]. To assess the effect of different shade signals on leaf senescence in soybean, we conducted experiments with simulated shade regimes of low R:FR, LBL (reduced blue light), and low R:FR+ LBL. Briefly, one unifoliate leaf of a seedling was covered with two layers of yellow filters to mimic the LBL condition, while another unifoliate leaf on the opposite side of the same seedling was covered with two layers of transparent filters and served as the control (Fig. 1a). For the low R:FR regime, far-red light was supplemented to one unifoliate leaf of a seedling, and the opposite leaf was used as the control. Phenotypic analysis of the leaves under various shade regimes revealed significantly accelerated senescence of the LBL-treated leaves (Fig. 1b), with lower chlorophyll content and higher expression levels of senescence marker genes *GmSAG12*, *GmSAG13*, and *GmSAG113* compared to the control (Fig. 1c, d). The unifoliate leaves under low R:FR+LBL conditions displayed even more marked senescence with lower chlorophyll content than those under low R:FR or LBL conditions (Supplementary Fig. 1). These results suggest that LBL is an essential shade signal promoting leaf senescence in soybean, and that it operates through an independent pathway with low R:FR to induce leaf senescence.

### GmCRY1s negatively regulate LBL-induced leaf senescence in soybean

Next, we investigated whether GmCRYs are involved in mediating light-induced leaf senescence (LBL) in soybean. The soybean genome encodes four *CRY1* (*GmCRY1a-1d*) and three *CRY2* (*GmCRY2a-2c*) co-orthologous genes[47]. We previously generated the *Gmcry1abcd* quadruple mutant (*Gmcry1s-qm*) and the *Gmcry2abc* triple mutant (*Gmcry2s-tm*) using CRISPR-Cas9 technology[44]. Phenotypic analysis revealed that both the *Gmcry1s-qm* and *Gmcry2s-tm* mutants displayed earlier leaf senescence compared to the wild-type under long-day and natural field conditions (Supplementary Figs. 2a, 3). The *Gmcry1s-qm* mutant displayed a more prominent decrease in chlorophyll content, higher cotyledon and leaf senescence index, and higher expression levels of senescence marker genes (*GmSAG12*, *GmSAG13* and *GmSAG113*) than the *Gmcry2s-tm* mutant (Supplementary Fig. 2b–e), indicating that GmCRY1s play a dominant role in controlling leaf senescence in soybean. This was further supported by the observation that the *GmCRY1b-OE* lines demonstrated delayed leaf senescence phenotype (Supplementary Figs. 4–6). To avoid the effect of different plant status on leaf senescence[44], we examined the senescence phenotype using detached leaves of the *Gmcry1s-qm* mutant, *GmCRY1b-OE* lines and wild-type plants. The results again showed that the *Gmcry1s-qm* mutant senesced faster, whereas *GmCRY1b-OE* lines senesced slower than wild type (Supplementary Fig. 7). These findings underscore the notion that GmCRY1b is a negative regulator of leaf senescence in soybean.

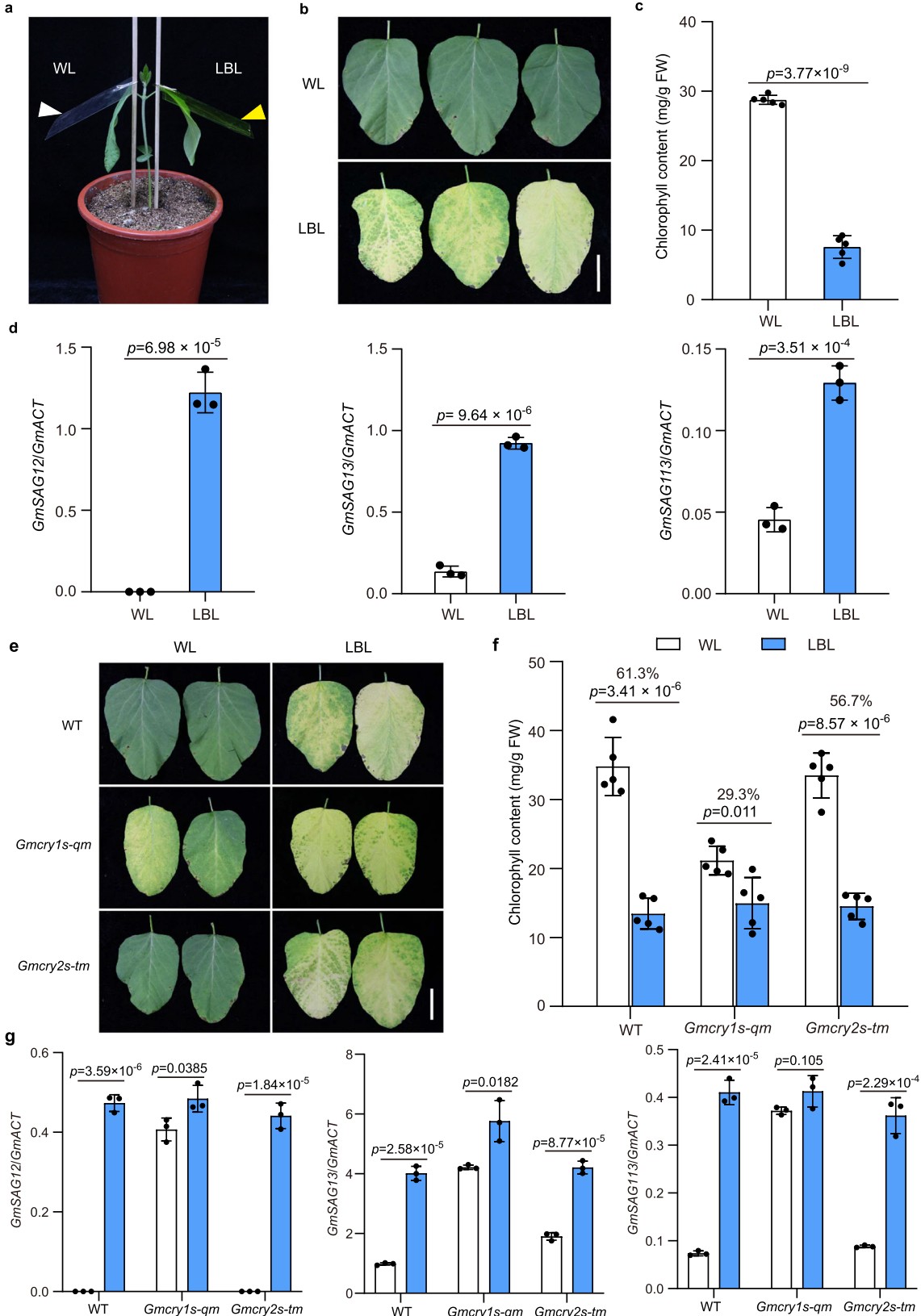

To assess the role of GmCRYs in LBL-induced leaf senescence, we compared the performance of the *Gmcry1s-qm* and *Gmcry2s-tm* mutants under these conditions. Our findings revealed that LBL-induced leaf senescence was significantly disrupted by *Gmcry1s* mutations and slightly impaired by *Gmcry2s* mutations (Fig. 1e–g), indicating that GmCRY1s are dominantly responsible for LBL-induced leaf senescence in soybean. Furthermore, the LBL-induced leaf senescence phenotype was less pronounced in the *GmCRY1b-OE* lines than in the wild-type plants (Supplementary Fig. 8), suggesting that overexpression of *GmCRY1b-YFP* may be utilized to improve the leaf senescence trait under dense planting or intercropping conditions.

**Fig. 1 | Phenotypic analysis of LBL-induced leaf senescence in the wild-type TL1 and *Gmcry* mutants. a** Experimental scheme for LBL treatment. For a pair of unifoliate leaves, one was covered with two layers of yellow filter to imitate the LBL condition, and the other one was covered with two layers of transparent filters as the control. White and yellow arrows indicate the transparent and yellow filters, respectively. **b** Leaf senescence phenotypes of wild-type TL1 cultivar induced by LBL treatment. Seedlings were de-etiolated under continuous white light for 10 days, then a pair of unifoliate leaves were treated with different light regimes (LBL or WL) for 14 days as in (**a**). Scale bar, 5 cm. **c** Chlorophyll content in the leaves as in (**b**). Values are means ± SD (*n* = 5 biological replicates). **d** Relative transcript levels of senescence-associated genes *GmSAG12*, *GmSAG13*, and *GmSAG113* in the leaves as in

(**b**). Values are means ± SD (*n* = 3 biological replicates). The *GmActin* gene was used as the internal control. **e** Leaf senescence phenotypes of indicated lines treated by WL and LBL as in (**b**). Scale bar, 3 cm. **f** Chlorophyll content in the leaves as in (**e**). The percentage decrease in chlorophyll content under WL compared to LBL is indicated by the values above the respective *p* values. Values are means ± SD (*n* = 5 biological replicates). **g** Relative transcript levels of senescence-associated genes in the indicated lines in response to LBL treatment as in (**b**). The unifoliate leaves were collected for RT-qPCR analysis. Values are means ± SD (*n* = 3 biological replicates). The *GmActin* gene was used as the internal control. All above *P* values were calculated by unpaired two-tailed *t*-test. Source data are provided as a Source Data file.

## GmCRY1b interacts with DELLA proteins RGAa and RGAb in response to blue light

To elucidate the molecular mechanism via which GmCRY1s repress leaf senescence, we used a yeast two-hybrid (Y2H) system to identify its potential interaction partners. Considering the strong autoactivation ability of GmCRY1b, the C-terminal truncated form of GmCRY1b (GmCCT1b-33aa) was fused with the BD domain of the pBridge vector to obtain the bait for yeast hybrid screening (Fig. 2a). The yeast cells harboring the bait were transformed with a library of prey protein-encoding cDNAs fused to GAL4-AD. The screening discovered eight putative interacting partners of GmCRY1b (Supplementary Data 2), including a DELLA protein GmRGAa (Glyma.05G140400), which has a paralogous protein GmRGAb (Glyma.08G095800, with 95.4% similarity to GmRGAa) in soybean (Supplementary Fig. 9). Further, the open reading frames (ORF) of GmRGAa and GmRGAb in soybean were fused to the AD domain of the pGADT7 vector and used for further interaction experiments with various truncated versions of GmCRY1b. The bait and prey vectors were co-transformed into yeast, and the protein-protein interactions were reconstructed. The two-hybrid screening demonstrated that a region of 33 amino acids (451-483) in the middle part of GmCRY1b is essential to interact with the DELLA proteins physically (Fig. 2b, c, and Supplementary Fig. 10). To determine the motif of DELLA protein that interacts with GmCRY1b, we used truncated forms of GmRGAa (N-terminal and C-terminal) for Y2H experiments. Our results showed that the N-terminal truncated form of GmRGAa caused strong auto-activation in yeast, whereas the C-terminal truncated form of GmRGAa physically interacted with GmCRY1b, as shown in Supplementary Fig. 11.

We further investigated the interaction between GmCRY1b and DELLA proteins in plant cells by infiltrating *Agrobacterium* expressing indicated proteins into soybean leaves[48]. The results revealed that GmCRY1b interacted with GmRGAa and GmRGAb in a blue light-dependent manner, as evidenced by the co-immunoprecipitation (Co-IP) assay (Fig. 2d). This blue light-dependent interaction between GmCRY1b and DELLA proteins was further corroborated using the bimolecular fluorescence complementation (BiFC) assay in soybean mesophyll protoplasts (Fig. 2e, Supplementary Fig. 12), and a split-LUC assay in *Nicotiana benthamiana* leaves (Supplementary Fig. 13), with the interaction being abrogated under LBL conditions. The BiFC signals were observed mainly in the nucleus, which is consistent with the subcellular localization results that showed GmCRY1b occupies the same location as DELLA proteins in the nucleus (Supplementary Fig. 14), suggesting a function of the GmCRY1b-GmDELLA complex in the nucleus.

## Photoactivated GmCRY1b inhibits the degradation of RGAa and RGAb

Based on the knowledge that DELLA proteins act as central repressors in gibberellin (GA) signaling pathway[31], we tested if GA acts as a senescence-associated hormone in soybean. We conducted experiments using GA_3 or PAC (paclobutrazol) treatment on seedlings, and our results showed that GA_3 accelerated leaf senescence, while PAC

delayed it, with higher chlorophyll content and lower senescence index (Supplementary Fig. 15). Furthermore, overexpression of *GmGA2ox-7a*, which deactivates bioactive GAs[44], also show delayed senescence (Supplementary Fig. 16), indicating that GA is a senescence-promoting hormone in soybean.

We then investigated the role of GmCRY1s in GA-mediated leaf senescence and found that GA could induce significant senescence in *Gmcry1s-qm* mutant (Supplementary Fig. 17), suggesting that GmCRY1s inhibit GA-mediated leaf senescence in soybean. Given that GA promotes the degradation of DELLA proteins, we hypothesized that binding of GmCRY1b to DELLA proteins could impede access of the GA receptor GID1 to DELLA proteins, thus interfering with DELLA degradation. To test this possibility, we used the RICE system to compare the protein levels of GmRGAa and GmRGAb in the *Gmcry1s-qm* mutant, *GmCRY1b-OE* line and wild type under continuous light or after LBL treatment. The immunoblot results revealed that GmRGAa and GmRGAb proteins in the wild-type callus were gradually reduced under LBL conditions at 3, 5, and 8 h, while the proteins remained constant under continuous white light (Supplementary Fig. 18). Consistent with the results observed in soybean hairy root callus, GmRGAb protein also showed a similar reduction in response to LBL in the *Flag-GmRGAb-1* stable transgenic line (Fig. 2f, g). Moreover, the GmRGAa and GmRGAb protein levels were significantly lower in the *Gmcry1s-qm* mutant but markedly higher in the *GmCRY1b* overexpressing line in comparison to the wild type (Fig. 2h–j). These results demonstrate that GmCRY1b stabilizes GmRGAa and GmRGAb in a blue light-dependent manner and releases the degradation of GmRGAa and GmRGAb in response to LBL.

## GmRGAa and GmRGAb negatively regulate LBL-induced leaf senescence

To determine the roles of DELLA proteins in regulating soybean leaf senescence, we knocked out the *GmRGAa* and *GmRGAb* genes using CRISPR-Cas9 technology. Multiple mutants were identified for each gene, including *Gmrgaa-1*, *Gmrgab-1*, and *Gmrgab-2* (Supplementary Fig. 19) with mutations creating premature stop codons in the targeted genes (Supplementary Fig. 20). We then crossed *Gmrgaa-1* with *Gmrgab-1* to obtain the *Gmrgaa Gmrgab* double (*Gmrgas-dm*) mutant. Although the *Gmrgaa-1*, *Gmrgab-1*, and *Gmrgab-2* single mutants did not exhibit an obvious senescence phenotype, the *Gmrgas-dm* mutant displayed a significantly precocious leaf senescence with lower chlorophyll content, higher cotyledon and leaf senescence index, and higher expression levels of *SAGs* compared to the wild type (Supplementary Fig. 21). Additionally, two *GmRGAb* overexpression lines (*GmRGAb-OE-1* and *GmRGAb-OE-2*) harboring the *GmRGAb* coding sequence driven by the 35S promoter were generated (Supplementary Fig. 22). Compared to the wild type, the *Gmrgas-dm* mutant presented premature leaf senescence, while the *GmRGAb-OE* lines showed delayed leaf senescence under natural field conditions (Supplementary Fig. 23), indicating that GmRGAa and GmRGAb play a negative role in leaf senescence. We further analyzed the agronomic traits of transgenic lines lacking or overexpressing *GmRGAs* at the R8 stage under natural field conditions. Our results demonstrated that the main

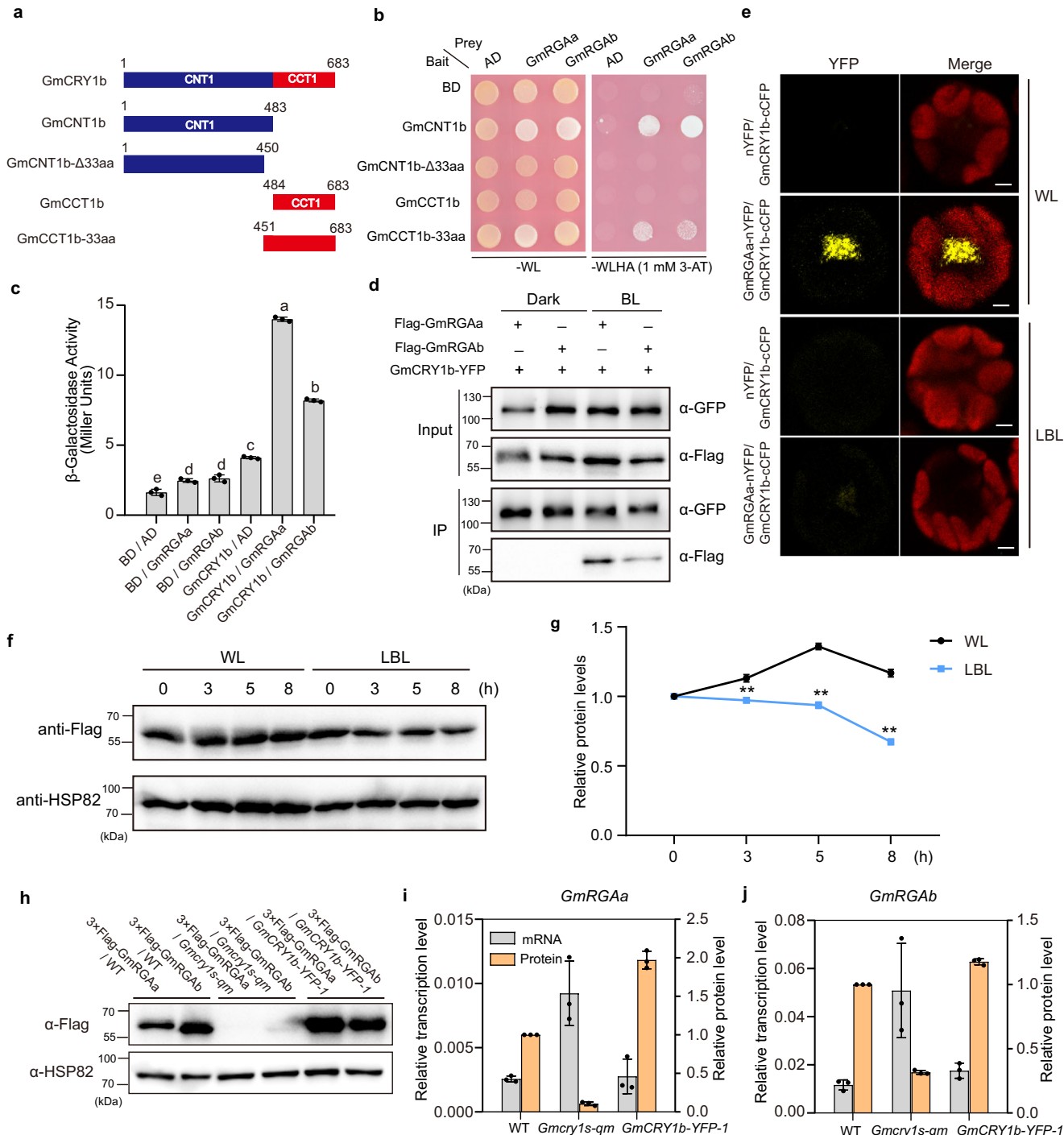

**Fig. 2 | Evaluation of the interaction between GmCRY1b and DELLA proteins.**
**a** Schemes display full-length and truncated versions of the GmCRY1b protein.
**b** Yeast two-hybrid (Y2H) assays show the interaction of the different truncated versions of GmCRY1b with DELLA proteins. SD-WL, minimal medium lacking Trp and Leu; SD-WLHA, selective medium lacking Trp, Leu, His, and adenine, and supplemented with 1 mM 3-AT. **c** Quantitation of β-galactosidase activity (Miller units) for each pair of bait and prey proteins as indicated by liquid assays. Values are mean ± SD ($n = 3$ biological replicates). The lowercase letters indicate significant differences (One-way ANOVA with two-sided Tukey test at a significance level of 0.05). **d** Co-IP assays demonstrate the blue light-dependent interaction of GmCRY1b with GmRGAa and GmRGAb in soybean leaves. Dark-adapted soybean leaves were either kept in the dark or exposed to blue light (50 μmol m$^{-2}$ s$^{-1}$) for 3 h. **e** BiFC analysis of protein interactions between GmCRY1b and GmRGAa in soybean protoplasts under WL or LBL conditions. The YFP fluorescence is shown, with the

Merge representing the combination with the fluorescence of chloroplasts. Scale bars, 10 μM. The experiment was repeated three times with consistent results.
**f** Effects of LBL treatment on the abundance of GmRGAb. Seedlings were grown under continuous WL for 12 days, then transferred into LBL or kept in continuous WL. The first trifoliate leaves were collected at the indicated time points. **g** The relative proteins levels of Flag-GmRGAb, normalized to HSP82 as in (**f**), are shown. Values are mean ± SD ($n = 3$ biological replicates). Statistical significance was analyzed using a two-tailed Student's $t$-test, **$P < 0.01$, and $P = 2.03 \times 10^{-5}$, $5.57 \times 10^{-8}$ and $2.82 \times 10^{-8}$ at the time points of 3, 5 and 8 h. **h** Immunoblot analysis of transgenic GmRGAa and GmRGAb proteins in the indicated callus lines with anti-Flag antibody. **i, j** Relative transcript and protein levels of transgenic GmRGAa (**i**) and GmRGAb (**j**) in the indicated callus lines. Values are means ± SD ($n = 3$ biological replicates). Source data are provided as a Source Data file.

yield traits, such as node number, branch number, and total grain weight per plant, did not exhibit any significant change among each line (Supplementary Fig. 24).

To investigate the role of GmRGAa and GmRGAb in LBL-induced leaf senescence, we grew *Gmrgaa-1*, *Gmrgab-1* and *Gmrgas-dm* seedlings for LBL treatment (Supplementary Fig. 25a). Phenotypic analysis revealed that the reduction in chlorophyll content induced by LBL was gradually reduced in the *Gmrgaa-1*, *Gmrgab-1*, and *Gmrgas-dm* mutants compared to the wild type (Supplementary Fig. 25b). This, coupled with the fact that LBL promotes the degradation of GmRGAa and GmRGAb, suggests that LBL triggers leaf senescence at least partially through reducing the protein abundance of GmRGAa and GmRGAb in soybean.

### GmRGAa and GmRGAb are genetically downstream of GmCRY1s in regulating leaf senescence

To investigate the genetic relationship between GmCRY1s and DELLA proteins in modulating leaf senescence, we generated a *Gmrgas-dm/GmCRY1b-YFP-1* line by crossing the *Gmrgas-dm* mutant with the

*GmCRY1b-OE* line (*GmCRY1b-YFP-1*). The phenotypic assessment of these lines indicated that the *Gmrgas-dm* mutations partially suppressed the delayed leaf senescence caused by *GmCRY1b* overexpression. Compared to the *GmCRY1b-YFP-1* line, the *Gmrgas-dm/GmCRY1b-YFP-1* plants senesced faster but still significantly slower than the *Gmrgas-dm* mutant (Fig. 3a). Additionally, the *Gmrgas-dm/GmCRY1b-YFP-1* plants had a significantly lower chlorophyll content, higher cotyledon and leaf senescence index, and higher *SAGs* transcript levels (Fig. 3b–e) compared to the *GmCRY1b-YFP-1* plants at 25 days after sowing under long-day conditions. Further application of 10 μM GA₃ accelerated the process of leaf senescence in *GmCRY1b-YFP-1* and *Gmrgas-dm/GmCRY1b-YFP-1* plants to a similar extent to that as observed in the wild-type plants (Supplementary Fig. 26). This implies that other DELLA proteins may function alongside GmRGAa and GmRGAb, which are genetically downstream of GmCRY1s, to regulate leaf senescence in soybean.

### GmWRKY100 promotes leaf senescence in response to LBL

To further dissect the signaling pathway involved in GmCRY1s-mediated leaf senescence, we analyzed the differentially expressed

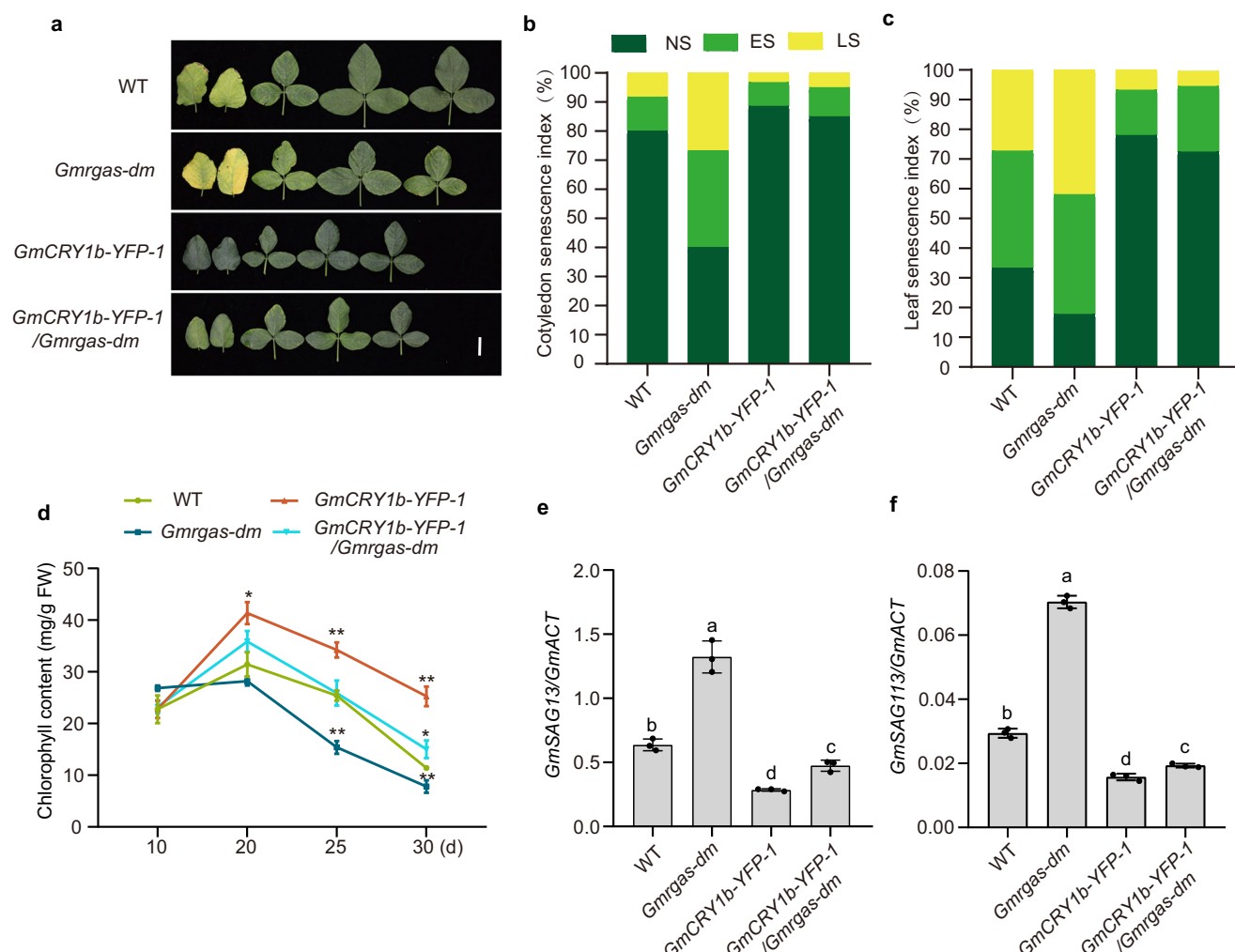

**Fig. 3 | GmCRY1b negatively regulates leaf senescence partially through the DELLA proteins. a** Leaf senescence phenotypes of wild-type TL1 cultivar, *Gmrgas-dm/GmCRY1b-YFP-1* transgenic plants and the corresponding parental lines under long-day conditions for 25 days. Scale bar, 5 cm. Cotyledon senescence index (**b**) and unifoliate leaf senescence index (**c**) as in (**a**). The cotyledon and unifoliate leaf senescence index was calculated at the age of 15 days and 25 days, respectively (*n* ≥ 20 biological replicates). **d** Chlorophyll content in the unifoliate leaf at the indicated leaf age as in (**a**). Values are means ± SD (*n* = 5 biologically independent plants), \**P* < 0.05, \*\**P* < 0.01 by two-tailed *t*-test. Relative transcript levels of senescence marker genes *GmSAG13* (**e**) and *GmSAG113* (**f**) in the unifoliate leaf at the leaf age of 25 days as in (**a**). Values are mean ± SD (*n* = 3 biological replicates). The lowercase letters indicate significant differences (One-way ANOVA with two-sided Tukey test at a significance level of 0.05). Source data are provided as a Source Data file.

genes (DEGs) among the wild type, *Gmcry1s-qm* and *GmCRY1b-OE* lines. We previously identified 3055 and 638 DEGs in the *Gmcry1s-qm* mutant and the *GmCRY1b-OE* line, respectively, compared to the wild type[44]. Among these, 249 genes exhibited opposite expression patterns in the *Gmcry1s-qm* mutant and *GmCRY1b-OE* line compared to wild type. We identified 14 senescence-related genes encoding WRKY TFs, MYB TFs, SEN4, cysteine protease, and B-BOX domain proteins that might

regulate leaf senescence in soybean (Supplementary Data 3). Among these, *Glyma.06G168400*, encoding a typical WRKY TF named GmWRKY100 in a previous study[49] (Supplementary Fig. 27a), exhibited significantly down-regulated expression in the *GmCRY1b-OE* line and up-regulated expression in the *Gmcry1s-qm* mutant (Fig. 4a, b), suggesting that GmWRKY100 may play a role in GmCRY1s-mediated regulation of leaf senescence.

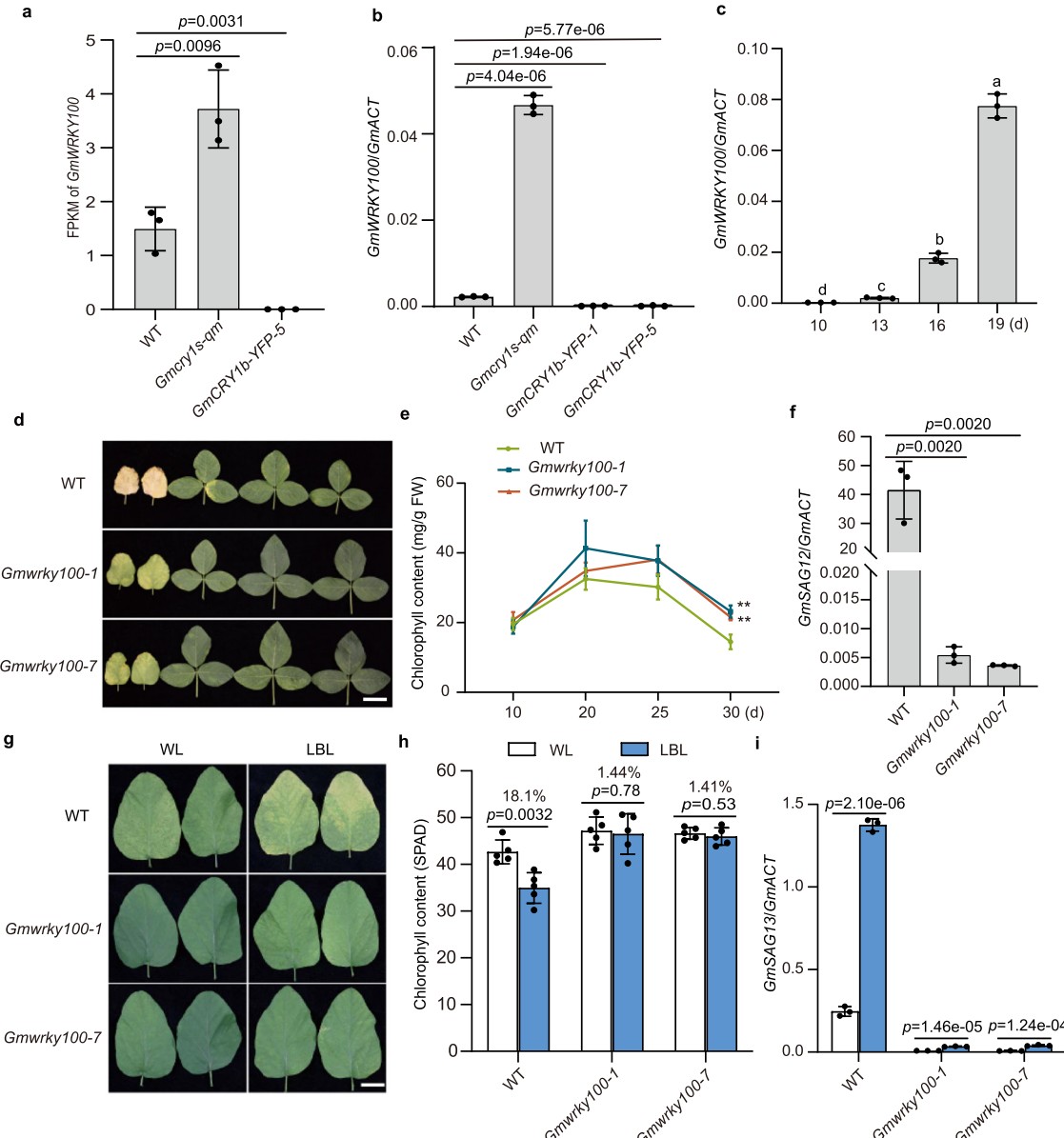

**Fig. 4 | *GmWRKY100* is a senescence enhancer and mediates LBL-induced leaf senescence in soybean. a** Normalized mRNA expression levels (FPKM) of *GmWRKY100* in the indicated genotypes. Values are means ± SD (*n* = 3 biological replicates). **b** Relative transcript levels of *GmWRKY100* in the indicated genotypes under long-day conditions for 12 days. Values are means ± SD (*n* = 3 biological replicates). **c** Relative transcript levels of *GmWRKY100* in wild-type TL1 cultivar at the indicated leaf age under long-day conditions. Values are means ± SD (*n* = 3 biological replicates). Different lowercase letters indicate significant differences (One-way ANOVA with two-sided Tukey test at a significance level of 0.05). **d** Leaf senescence phenotypes of the indicated genotypes under long-day conditions for 35 days. Scale bar, 5 cm. **e** Chlorophyll content at the indicated leaf age as in (**d**). Values are means ± SD (*n* = 5 biologically independent plants), **\**P* < 0.01. **f** Relative transcript levels of *GmSAG12* at the indicated leaf age 25 days. Values are means ± SD (*n* = 3 biological replicates). **g** Leaf senescence phenotypes of *Gmwrky100*

mutants and TL1 under WL or LBL conditions. Seedlings were grown under continuous white light for 10 days, then a pair of unifoliate leaves were treated with the different light regimes for 14 days. Scale bar, 3 cm. **h** Chlorophyll content (Measured as SPAD) of the unifoliate leaf in the indicated lines under WL and LBL conditions. Seedlings were grown under long-day conditions for 10 days, then a pair of unifoliate leaves were treated with LBL or WL for 12 days. Values are means ± SD (*n* = 5 biologically independent plants). **i** Relative transcript levels of *GmSAG13* in the unifoliate leaves of indicated lines in response to LBL treatment as in (**g**). Values are means ± SD (*n* = 3 biological replicates). The *GmActin* gene was used as the internal control. All above *P* values were calculated by unpaired two-tailed *t*-test. Source data as well as the results of two-way ANOVA statistical analysis for significant interaction between genotypes and light treatments are provided in the Source Data file.

The transcriptional level of *GmWRKY100* was observed to increase in association with the onset of leaf senescence (Fig. 4c), suggesting that *GmWRKY100* promotes leaf senescence in soybean. Notably, the expression levels of *GmCRY1b*, *GmRGAa* and *GmRGAb* also increased as leaf senescence progressed (Supplementary Fig. 28), suggesting the existence of negative feedback regulation among GmCRY1s, DELLAs, and GmWRKY100 in the process of leaf senescence. To test this, we generated *GmWRKY100* knockout mutants using the CRISPR/Cas9-engineered genome-editing approach. We identified two independent mutants, *Gmwrky100-1* and *Gmwrky100-7*, with 2 bp and 5 bp deletions, respectively, which caused frame-shift mutations and premature stop codons (Supplementary Fig. 27b, c). Compared with the wild type, these mutants showed delayed leaf senescence under green-house and natural field conditions (Fig. 4d, Supplementary Fig. 29), with slower chlorophyll degradation (Fig. 4e), lower cotyledon and leaf senescence index (Supplementary Fig. 30), and lower expression levels of *GmSAG12* (Fig. 4f). LBL treatment significantly induced the expression of *GmWRKY100*, whereas its expression remained constant under continuous white light (Supplementary Fig. 31). To determine the function of GmWRKY100 in LBL-induced leaf senescence, we grew wild-type plants and *Gmwrky100* mutants under WL and LBL conditions, respectively. Phenotypic analysis revealed a reduced LBL-induced leaf senescence in the *Gmwrky100* mutant compared to the wild type (Fig. 4g, Supplementary Fig. 32). The statistical analysis of two-way ANOVA indicated a significant interaction between genotypes and light treatments in terms of chlorophyll content and senescence marker gene *GmSAG13* (Fig. 4h, i). These results suggest that the *GmWRKY100* gene plays a significant role in LBL-induced leaf senescence.

### LBL releases the inhibitory effect of DELLA proteins on *GmWRKY100* transcription

Given the fact that *GmRGAa* and *GmRGAb* delay leaf senescence, while *GmWRKY100* promotes it, we surmise that *GmWRKY100* transcription may be negatively regulated by GmRGAa and GmRGAb. This is supported by the observation that the mRNA level of *GmWRKY100* was upregulated by more than 20-fold in *Gmrgas-dm* mutants compared to wild type (Fig. 5a). Furthermore, treatment with 10 µM GA$_3$ significantly increased the mRNA levels of *GmWRKY100*. However, the GA-induced *GmWRKY100* expression is disrupted in *Gmcry1s-qm* mutant (Fig. 5b, Supplementary Fig. 33). On the other hand, over-expression of *GmGA2ox7a*, which deactivates GA, resulted in a down-regulation of *GmWRKY100* transcription (Fig. 5c), suggesting that DELLA proteins can inhibit *GmWRKY100* expression.

To further investigate whether DELLA proteins can directly regulate *GmWRKY100* transcription, we conducted a dual-luciferase (LUC) reporter assay by transiently expressing the *p35S:GUS*, *p35S:GmCRY1b*, *p35S:GmRGAa*, or *p35S:GmRGAb* constructs with the *ProGmWRKY100:LUC* construct in soybean mesophyll protoplasts (Fig. 5d). The results demonstrate that GmRGAa and GmRGAb effectively suppressed the relative LUC activity compared to the GUS control protein under white light (Fig. 5e). To confirm this further, a chromatin immunoprecipitation (ChIP)-qPCR assay was performed, which revealed that GmRGAb was associated with the *GmWRKY100* promoter around the AE-box (part of a module for light response), at 337 bp upstream of ATG (Fig. 5f, g). These results indicate that DELLA proteins can directly repress the transcription of *GmWRKY100*.

Interestingly, the inhibitory effect of DELLA proteins on the transcriptional activity of *GmWRKY100* was reduced under LBL conditions (Fig. 5e). These results together suggest a working model in which LBL triggers the degradation of DELLA proteins, resulting in increased transcription of *GmWRKY100* and accelerated leaf senescence in soybean (Fig. 5h). Since DELLA proteins lack a canonical DNA-binding domain, it is likely that unidentified transcriptional factors play a role in regulating the activity of the *GmWRKY100* promoter,

while DELLA proteins act as repressors by interacting with these factors. To be noted, adding GmCRY1b made no observable impact on the transient expression assays (Fig. 5e), likely indicating a sufficient presence of CRY1 in the protoplasts.

Furthermore, the *Gmwrky100* mutant lines showed no significant phenotypic penalties and exhibited potential for yield improvement under natural field conditions (Fig. 6). An examination of the agronomic traits of the two mutant lines revealed an 11% increase in pod number per plant and a 10% increase in grain weight per plant, with no changes in grains size and 100 seed weight, compared to the wild-type control (Fig. 6g, h, and Supplementary Figs. 34, 35). Additionally, the growth stages showed no significant differences between the wild-type plants and *Gmwrky100* mutants (Supplementary Fig. 36), indicating that the increased grain yield in the *Gmwrky100* mutants may have been a result of delayed senescence and extended photosynthesis period in soybean.

## Discussion

Shading caused by the upper leaves or neighboring canopies triggers early leaf senescence in lower positions, which adversely affects crop yield potential and stability[23,50]. However, the mechanism driving this phenomenon has not been fully elucidated. In this study, we demonstrated that lower blue light levels under shaded conditions effectively induce premature leaf senescence in soybean. Further, we identified a GmCRY1s-DELLA-GmWRKY100 module that functions in LBL-induced leaf senescence in soybean. Our findings provide insight into how plants perceive and interpret LBL signals when shaded by neighboring plants to regulate the progression of leaf senescence.

It has long been established that low R:FR ratio can induce leaf senescence in model plant *Arabidopsis*, but the roles of blue light and cryptochrome in this process have not been as well-studied. This is because the *cry1cry2* mutant does not typically show altered leaf senescence phenotypes compared to the wild type under normal growth conditions or when using detached leaves incubated in the dark[21]. In contrast, soybean leaves of the *Gmcry1s-qm* mutant and *Gmcry2-tm* mutant showed different extents but clear premature senescence phenotypes under both natural and detached conditions (Supplementary Figs. 2, 7). Notably, a recent study observed that the *cry1cry2* mutant displayed a premature leaf senescence phenotype when the detached leaves were subjected to an FR light pulse prior to incubation under blue light[49]. Additionally, CRY2 appears to play a more prominent role than CRY1 in mediating blue light inhibition of leaf senescence in *Arabidopsis*, which contrasts with the observation that GmCRY1s, but not GmCRY2s, predominantly regulates leaf senescence in soybean. All of these observations suggest that cryptochromes play a conserved role in regulating leaf senescence but have evolved with distinct mechanisms in different plant species.

Previously, we reported that GmCRY2 may delay leaf senescence by inhibiting the transcriptional activation activity of GmCIB1 in soybean[43]. However, the *cib1cib2cib3cib4cib5* quintuple mutant of *Arabidopsis* did not show any obviously altered leaf senescence phenotype. Instead, HY5 and PIF4/5 were demonstrated to be downstream of CRY2, suppressing and enhancing leaf senescence, respectively[49]. Here, we further established a GmCRY1-DELLA-WRKY100 regulatory cascade in modulating LBL-induced leaf senescence in soybean. Further studies are needed to investigate whether HY5 and PIF4/5 homologous proteins also participate in GmCRY1-mediated leaf senescence process in response to the variation in blue light fluence rate in soybean.

DELLA proteins are central components in the control of plant growth responses to adapt to environmental changes. Recently, the physical interactions between CRY1 and DELLA proteins have also been reported in *Arabidopsis* and wheat, suggesting that the CRY-DELLA signaling module is conserved among various species[51–53]. Here, we found that GmCRY1b also interacts with GmRGAs (GmRGAa and

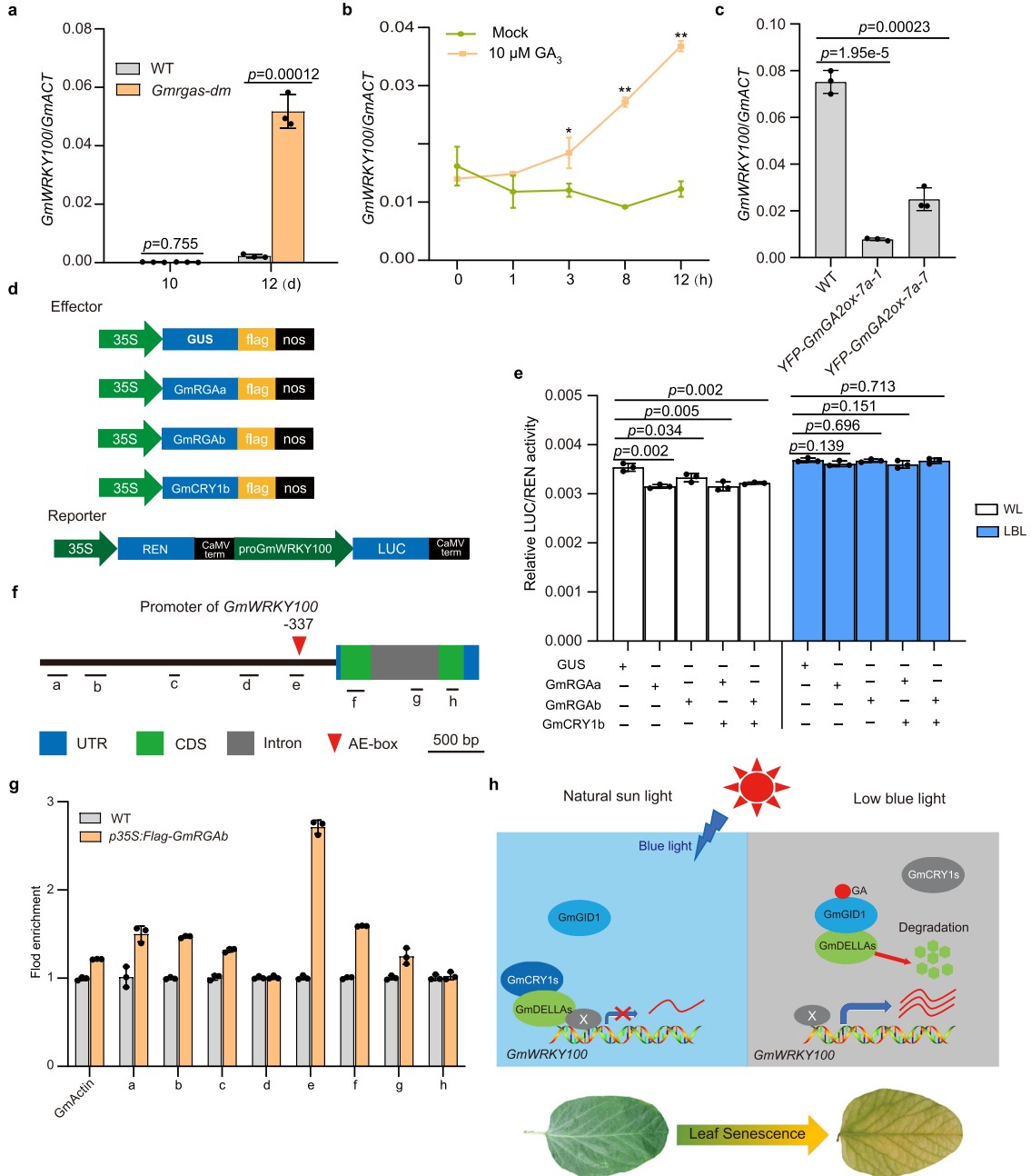

**Fig. 5 | DELLA proteins GmRGAa and GmRGAb associate with the promoter of *GmWRKY100* and suppress its transcription. a** Relative expression levels of *GmWRKY100* in the unifoliate leaves of wild-type TL1 cultivar and *Gmrgas-dm* plants under long-day conditions for indicated days. Values are relative to the control gene *GmACT* and represent means ± SD (*n* = 3 biological replicates). **b** Relative expression levels of *GmWRKY100* in the unifoliate leaves of 14-day-old seedlings treated with 10 μM GA$_3$ for the indicated time. Values are means ± SD (*n* = 3 biological replicates), **$P$ < 0.01. **c** Relative expression levels of *GmWRKY100* in the indicated lines grown under continuous light conditions for 15 days. Values are means ± SD (*n* = 3 biological replicates). **d** Constructs of GmCRY1b, GmRGAa, and GmRGAb were used for transient dual-luciferase reporter assay in soybean protoplast. **e** LUC and REN activity were measured after culturing the protoplasts under continuous white light or LBL conditions for 4 h. Values are means ± SD (*n* = 3 biological replicates). All above *P* values were calculated by unpaired two-tailed *t*-test. **f** Schematic diagram of *GmWRKY100* gene and regions tested for enrichment

by ChIP assay. The filled red arrowhead represents the AE-box (part of a module for light response). **g** ChIP analysis of the interaction between GmRGAb and the *GmWRKY100* chromatin regions. Values are means ± SD (*n* = 3 biological repli-cates). **h** A working model of GmCRY1s-GmDELLAs-GmWRKY100 signaling path-way in regulating leaf senescence in soybean. Under LBL conditions, GA triggers the degradation of DELLA proteins through the 26S proteasome pathway, and the expression of *GmWRKY100* is released from the repression of DELLA proteins, which further promotes the expression of *GmSAGs* and leaf senescence. Upon blue light illumination, GmCRY1s were activated and physically interacted with GmRGAa and GmRGAb and to enhance their protein stability. Meanwhile, DELLA proteins could bind to the promoter of *GmWRKY100* with unknown factors and repress its expression and the process of leaf senescence. Thick and thin blue arrows denote the expression of *GmWRKY100* being induced and repressed, respectively. Source data are provided as a Source Data file.

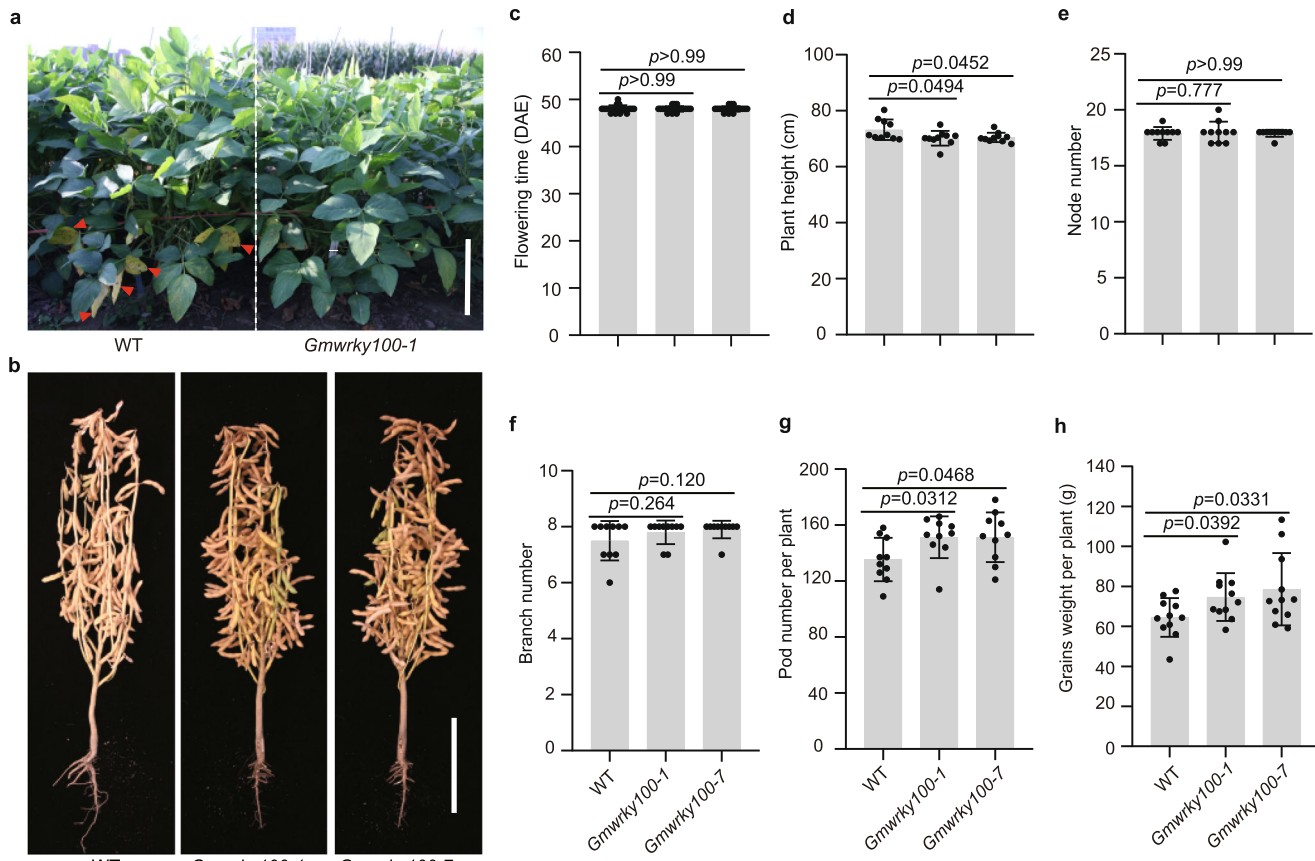

**Fig. 6 | Phenotypic and yield trait characteristics of *Gmwrky100* mutants in the field. a** Representative image of wild-type and *GmWRKY100* mutants grown under natural field conditions at the age of 97 days after sowing. The full red arrows indicate senescent leaves. Scale bar, 20 cm. **b** Representative images of the indicated lines grown under natural field conditions at the R8 stage. Scale bar, 20 cm. Phenotypic comparison between TL1 and *Gmwrky100* mutation lines in flowering time (**c**) (*n* = 18 biologically independent plants), plant height (**d**), node number (**e**), branch number (**f**), pod number per plant (**g**), and grain weight per plant (**h**) (*n* = 10 biologically independent plants). Values are means ± SD. All above *P* values were calculated by unpaired two-tailed *t*-test. Source data are provided as a Source Data file.

GmRGAb) in a blue light-dependent manner in soybean (Fig. 2d), and loss-of-function of GmCRY1s largely abolishes the accumulation of GmRGAs (Fig. 2h–j). Furthermore, LBL reduces the activity of GmCRY1s and the protein abundance of GmRGAs. We speculate that GmCRY1s may protect GmRGAs through two potential mechanisms. Firstly, GmCRY1s may protect GmRGAs by interfering with COP1[54], a component of multimeric E3 ligases that physically interacts with RGA[54], targeting it to degradation in the proteasome[55]. Second, GmCRY1s may affect the levels of active gibberellins[56], and hence the abundance of RGA via the canonical GA pathway. Future experiments will be necessary to discriminate between the two possibilities.

It has been proposed that delaying leaf senescence in soybean could potentially lead to higher yield[57]. However, the outcome of this practice has been controversial. For example, the 'stay-green' genotype *GGd1d1d2d2* increased seed yield in growth chamber conditions but reduced yield outdoors compared to its near-isogenic line wild type 'Clark'[58]. In contrast, a double mutant of *SGR1* and *SGR2*, which is a typical 'stay-green' variety (Z1), showed a higher yield performance compared to its parents JD74[59]. Here, we showed that knocking out *GmWRKY100* can increase yield per plant by more than 10%, potentially by increasing the rate of photosynthesis and extending the photosynthetically functional phase without obvious developmental defects. Collectively, this work provides a promising perspective to engineer the components of the light signaling pathway and obtain crops with ideal leaf senescence ideotypes to boost crop grain yield.

## Methods

### Plant materials and growth conditions
All genetically modified lines were constructed with soybean cultivar (*Glycine max* 'Tian-Long 1') mediated by *Agrobacterium tumefaciens* (EHA105 strain) for stable transformation. The elite soybean cultivar Tian-Long 1 was used as the WT control in this study, which was provided by the Oil Crops Research Institute of the Chinese Academy of Agricultural Sciences. All soybean plants were grown under long-day conditions (16 h light/8 h dark) or short-day conditions (8 h light/16 h dark) at 26 °C with light illumination in the green-house for phenotypic investigation. *Nicotiana benthamiana* plants were grown under short-day (10 h light/14 h dark) conditions at 26 °C in the green-house. Light was provided by the GreenPower light-emitting diode (LED) toplighting (HiPoint Brand), LBL light treatment was achieved by filtration of white light LED through two layers of yellow filter#101 (Lee Filters, CA) as previously described[15,60]. HiPoint HR-350 spectrometer was used to measure the quality and intensity of light.

### Accession numbers, vector construction, and soybean transformation
Gene sequence data in this article can be obtained from the Phytozome database (https://phytozome.jgi.doe.gov/) under the following accession numbers: *GmCRY1a* (Glyma.04G101500), *GmCRY1b* (Glyma.06G103200), *GmCRY1c* (Glyma.14G174200), *GmCRY1d* (Glyma.13G089200), *GmRGAa* (Glyma.05G140400), *GmRGAb* (Glyma.08G095800), *GmWRKY100* (Glyma.06G168400), *GmSAG12*

(*Glyma.11G144900*), *GmSAG13* (*Glyma.12G059200*), *GmSAG113* (*Glyma.14G066400*), *GmGA2ox7a* (*Glyma.20G141200*), and *GmActin* (*Glyma.18G290800*).

To construct the overexpression plant transformation vectors, the coding DNA sequence (CDS) of each indicated gene was amplified by PCR using cDNA derived from young leaves of Williams 82 seedlings, cloned into the gateway entry vector pDONR-Zeo by BP reaction, and then further cloned into the destination binary overexpression vector pEarleyGate101 or pEarleyGate104 by LR reaction using the Gateway recombination system (Invitrogen) following the manufacturer's instructions[61]. To generate the CRISPR-Cas9-engineered mutants, gRNAs were designed using the web tool CRISPRdirect (http://crispr.dbcls.jp/)[62]. The efficiency of each candidate gRNA was estimated using the soybean hairy root system[63], and efficient candidates were selected for soybean transformation. The above expression plasmids were individually introduced into *Agrobacterium tumefaciens* strain EHA105 via electroporation and then transformed into wild-type soybean (Tian-Long 1) by *agrobacterium*-mediated cotyledonary node transformation system[64]. Briefly, healthy seeds were selected and sterilized by chlorine for 18 h, then soaked into sterilized water overnight for imbibition. The seed coat was gently removed, and the swelled seeds were cut in half. The cotyledon explants were gently scratched at the cotyledon node and were immersed in *Agrobacterium* (EHA105) which harbors expression vectors for 30 min for infection. The infected explants were transferred to the co-cultured medium and subjected to dark conditions for three days at 25 °C. After 3 days of co-culture, the explants were washed with sterilized water supplemented with 50 mg/L timentin, 50 mg/L vancomycin, and 100 mg/L cefotaxime to remove the bacteria on the surface, then transferred to the shoot initiation medium with the hypocotyl embedded in the medium under a 12 h light/12 h dark photoperiod at 26 °C, and subcultured once for 10 days to fresh medium with three repetitions. The explants with tufted shoots were then transferred to shoot elongation medium and subcultured once for 10 days to fresh medium with three repetitions. The elongated shoots were cut and moved to the rooting medium. The shoot initiation medium and shoot elongation medium contain phosphinothricin (5 mg/L) to screen positive transgenic shoots.

## Total RNA isolation and gene expression analysis

To measure the expression of senescence-associated genes during leaf senescence, leaves of the indicated genotypes were flash-frozen in liquid nitrogen. Total RNA was extracted using TRIzol reagent (Invitrogen). The cDNA was synthesized from 2 μg of total RNA using Oligo(dT)$_{18}$ primer with TransScript II One-Step gDNA Removal and cDNA Synthesis SuperMix (TransGen). Real-time PCR instrument qTOWER 3G (analytikjena) was used for the quantitative PCR reaction following the manufacturer's instructions. Briefly, the cDNA was diluted 10-fold, and 5 μL of diluted cDNA was used as the template and amplified with Taq Pro Universal SYBR qPCR Master Mix (Vazyme) with specific primer sets (Supplementary Data 1) in a 20 μL quantitative PCR reaction, which was pre-denatured at 95 °C for 5 min, followed by a 40-cycle program (95 °C, 10 s; 60 °C, 20 s; 72 °C, 30 s per cycle). The soybean *GmActin* gene (*Glyma.18G290800*) was used as an internal control. The quantitative PCR results shown are the average (±SD) of three biological repeats. Primers used in the present study were listed in Supplementary Data 1.

## Western blot

To analyze the protein expression in transgenic plants and fairy root calluses, total proteins of Tian-Long 1, and the indicated transgenic plants were extracted with protein extraction buffer (50 mM Tris-HCl pH 7.5, 150 mM NaCl, 5 mM EDTA, 0.1% Triton X-100, 0.2% NP-40,1 mM PMSF and 1 tablet/50 mL of protease inhibitor cocktail). The homogenate was clarified by centrifugation at 14,000 g for 15 min at 4 °C, and aliquots of the supernatant were combined with 4×SDS sample loading buffer and heated at 99 °C for 8 min to denature the protein. The antibody anti-FLAG (M20008L) and anti-GFP antibody (598) were obtained from Abmart and MBL, respectively.

## Transient dual-luciferase reporter system

A 2.24-kb promoter sequence of *GmWRKY100* was amplified from soybean Williams 82 genomic DNA and inserted into the pGreenII 0800-LUC vector to control the luciferase (LUC) gene, which was used as reporter plasmids. The renilla luciferase (REN) gene under the control of the 35S promoter in the pGreenII 0800-LUC vector was used as an internal control[65]. The coding sequence of *GmCRY1b*, *GmRGAa*, *GmRGAb*, and GUS were amplified by PCR, inserted into the 0641-3×Flag vector, and used as an effector plasmid. 0641-GUS-3×Flag vector was set as negative effector control. Soybean mesophyll protoplasts were prepared, transfected, and cultured as described previously[66]. The ratio of LUC to REN was determined for the dual-luciferase reporter system (Promega, United States) on Centro XS³ LB 960 after culturing the protoplasts under normal white light or LBL conditions for 4 h. Transcriptional activity of the *GmWRKY100* promoter was calculated as LUC to REN ratio of three biological replicates.

## Measurement of chlorophyll content

The measurement of chlorophyll content was performed as described previously[43]. Briefly, 0.2 g of fresh sample of each indicated plant was frozen in liquid nitrogen, ground to powder, mixed thoroughly with 20 mL of 80% acetone, and stored at −20 °C for 1 h in the dark. Then the sample was centrifuged at 12,800 g for 3 min and 1 mL of supernatants was measured for absorbance at 663 nm and 645 nm. Chlorophyll concentrations were calculated using the following formulas:

$$\text{Concentration of total chlorophyll} = (20.2A_{645} + 8.02A_{663})\,\text{mg/g}$$

For the measurement of chlorophyll content in living plants, the SPAD value was scored using a SPAD meter (SPAD-502, Minolta Camera Co., Osaka, Japan) as previously described[67].

## Yeast two-hybrid assay

The yeast two-hybrid assay was performed as previously described[68]. In brief, the various truncated versions of GmCRY1b, and the full-length coding sequence (CDS) of DELLA proteins GmRGAa and GmRGAb were cloned into the bait vector pBridge and the prey vector pGADT7, respectively. The plasmids were transformed into the yeast strain AH109 (Clontech), and the yeast cells were grown on a minimal medium SD/-Leu-Trp according to the manufacturer's instructions (Clontech). Positive clones were selected on SD/-Ade-His-Leu-Trp selection medium containing 1 mM 3-AT (3-amino-1,2,4-triazole). Quantitation of β-galactosidase (β-gal) activity was determined as described by the manufacturer (Clontech).

## Root-induced callus expression system (RICE)

The *3×Flag-GmRGAa* and *3×Flag-GmRGAb* plasmids were introduced into *Agrobacterium tumefaciens* strain K599 via electroporation, which further infected the young seedlings of WT, *Gmcry1s-qm*, and *GmCRY1b* overexpressing line at the hypocotyl region to induce transgenic hairy roots according to previous methods with minor changes[69]. The positive transgenic hairy roots were screened in the callus induction medium (2.22 g/L Murashige & Skoog Basal Medium with Vitamins, 0.59 g/L MES monohydrate, 30 g/L sucrose, 1 mg/L 2, 4-D, 0.1 mg/L 6-BA, 0.1 g/L Timentin) contains 5 mg/L phosphinothricin (PPT). The positive transgenic calluses were cultured in the callus induction medium for 20 days under long-day (16 h of light /8 h of dark) conditions. These transgenic calluses lines were further confirmed by RT-qPCR and immunoblot analysis. Three independent hairy root calluses lines of each indicated genotype were used for transcriptional analysis and identification of protein levels.

## Soybean leaf injection

The seedlings of *GmCRY1b-YFP-1* transgenic line were grown under long-day conditions (16 h light/8 h dark) for seven days. The fully expanded unifoliate leaves were wiped with a brush[48]. The *Agrobacterium* strain GV3101 transformed with 3×Flag-GmRGAa or 3×Flag-GmRGAb overexpression vectors were resuspended with infiltration buffer (10 mM MES pH 5.6, 200 μM acetosyringone) to $OD_{600} = 1$, and then pressure-infiltrated into the lower epidermis of the leaves using a vacuum pump until the leaves were completely wet. The transformed soybean seedlings were recovered under continuous darkness for one day and then grown under normal long-day conditions.

## Co-immunoprecipitation assay

Soybean leaves of *GmCRY1b-YFP-1* overexpression line transiently transformed with indicated proteins by leaf injection were flash-frozen in liquid nitrogen, ground to powder, and mixed thoroughly with protein extraction buffer (50 mM Tris-HCl pH 7.5, 150 mM NaCl, 5 mM EDTA, 0.1% Triton X-100, 0.2% NP-40, 1 mM PMSF and 1 tablet/50 mL of protease inhibitor cocktail). The protein extracts were incubated at 4 °C for 30 min and centrifuged at 13,000 g for 30 min. After centrifugation, the supernatants were incubated at 4 °C with GFP-Trap Agarose (ChromoTek) for 4 h. The GFP-Trap Agarose was collected by spinning at 1500 rpm for 3 min and washed three times with the wash buffer (10 mM Tris-HCl pH = 7.5, 150 mM NaCl, 0.5 mM EDTA). The proteins were eluted from the GFP-Trap Agarose by mixing with 4×SDS-PAGE sample buffer, boiled for 8 min, and spun at 12,000 rpm for 5 min at room temperature, then subjected to immunoblot analysis. Immunoblots were performed using the anti-GFP antibody (MBL) for probing GmCRY1b-YFP and the anti-Flag antibody (Abmart) for probing Flag-GmRGAa and Flag-GmRGAb.

## Multiple alignment and construction of phylogenetic tree

The protein sequences of GAI(At1g14920), RGA(At2g01570), RGL1(At1g66350), RGL2(At3g03450), RGL3(At5g17490), WRKY45(At3g01970) and WRKY75(At5g13080) were retrieved from TAIR (https://www.arabidopsis.org/index.jsp). The DELLA proteins and GmWRKY100 protein sequences of Glycine max (GmRGAa, Glyma.05G140400; GmRGAb, Glyma.08G095800; GmWRKY100, Glyma.06G168400, and their homologous gene) are available at Phytozome (https://phytozome.jgi.doe.gov/pz/portal.html). Amino acid sequences of DELLA proteins, GmWRKY100, and their homologous proteins were aligned by ClustalW in MEGA X and manually adjusted. The phylogenetic tree was constructed using the neighbor-joining method in MEGA X software[70].

## Chromatin immuno-precipitation (ChIP) assay

Leaf samples were collected from 10-day-old seedlings under continuous light from Tian-Long 1, *p35S:3×Flag-GmRGAb* transgenic lines. ChIP assay was performed as previously described[71]. Briefly, 2 g leaves tissue sample was used in the ChIP experiment, samples were fixed on ice for 20 min in 1% formaldehyde under vacuum. Nuclei were isolated and sonicated. The solubilized chromatin was immunoprecipitated by anti-Flag M2 Magnetic Beads (M8823). The coimmunoprecipitated DNA was recovered and analyzed by RT-qPCR in triplicate. Relative fold enrichment was calculated by normalizing the amount of a target DNA fragment against that of a genomic fragment of a reference gene, *GmWRKY100* (*Glyma.06G168400*), and then by normalizing the value of the input DNA. The primers used for amplification are listed in (Supplementary Data 1).

## Split-luciferase assay

For split-LUC assays to detect protein-protein interactions, the cDNA fragments encoding GmCRY1b, GmRGAa, and GmRGAb were cloned into pCambia1300-nLUC and pCambia1300-cLUC. These constructs expressing Venus-nLUC, cLUC-Venus, GmRGAa-nLUC, GmRGAb-nLUC, and cLUC-GmCRY1b were introduced individually into *Agrobacterium* strain GV3101. The resulting colonies harboring the indicated constructs expressing nLUC or cLUC fusions were grown in LB medium overnight, collected by centrifugation, and resuspended in infiltration buffer (10 mM MgCl₂, 10 mM MES pH 5.6, 200 μM AS (Acetosyringone)). Bacterial suspensions were then mixed in a 1:1 ratio and infiltrated into *N. benthamiana* leaves. The infiltrated *N. benthamianas* were grown under white light or LBL conditions for 2 days after 12 h darkness-treatment. Then, *N. benthamiana* leaves were infiltrated with 1 mM D-luciferin sodium salt substrate and kept in the dark for 5 min. LUC signal was collected on a luminescent imaging workstation (Tanon 5200 Chemiluminescence imaging system).

## Bimolecular fluorescent complimentary (BIFC)

The CDS of GmCRY1b and DELLA proteins GmRGAa and GmRGAb were cloned into the pCCFP-GW or pNYFP-GW vector using a gateway recombination system. Soybean seedlings were grown under short-day (8 h light/16 h dark) conditions, at a light intensity of 120-180 μmol m⁻² s⁻¹ and a temperature of 26 °C. Mesophyll protoplasts were isolated from the unifoliate leaves of soybean and transformed following the reported procedure[72]. Protoplasts were transfected with the indicated plasmid DNA. Samples were incubated for 12 to 14 h in the dark at 26 °C, transferred to white light or LBL conditions for 2 h, and then analyzed under confocal microscopy (Zeiss LSM 980).

## Statistics and reproducibility

Multiple comparisons were conducted using GraphPad Prism 9.5 software with one-way or two-way ANOVA and two-sided Tukey test. For comparisons between two groups, two-tailed Student's *t*-tests were performed using Microsoft Excel. The statistical test employed and the corresponding number of individuals (*n*) for each experiment were both provided in the figure legends. For the expression analysis, at least three individual plants per tissue sample were pooled, and a minimum of three RT-qPCR reactions (technical replicates) were performed for three biological replicates. All experiments were conducted at least thrice for consistency.

## Reporting summary

Further information on research design is available in the Nature Portfolio Reporting Summary linked to this article.

# Data availability

All the data generated in this study are provided in the Supplementary Information and Source Data file. Source data are provided with this paper.

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

## Acknowledgements

This work was supported by the National Key Research and Development Plan (2023YFD1200600), the National Natural Science Foundation of China (32072091 and 31871705), the Innovation Program of Chinese Academy of Agricultural Sciences and the Central Public-Interest Scientific Institution Basal Research Fund to B.L., and the Sci-Tech Innovation 2030 (2022ZD0400701-2) to J.L.

## Author contributions

B.L. designed the research. Z.L., X.G.L., H.Y.L., Q.C.T., T.Z., and J.L. performed the experiments. Z.L., X.G.L., and H.Y.L. collected the phenotypic data. Z.L., X.G.L., and H.Y.L. analyzed data. B.L. and Z.L. wrote the manuscript.

## Competing interests

The authors declare no competing interests.
