## [Peer Review File · Nature Communications]

The mechanism of low blue light-induced leaf senescence
mediated by GmCRY1s in soybeanReviewer #1 (Remarks to the Author):

The manuscript by Li et al. reported the possible molecular mechanisms of GmCRY1s, The blue light receptors, in regulating low blue light-induced leaf senescence in soybean. They provided molecular and genetic evidence to demonstrate that GmCRY1s interact with DELLA proteins under light-activated conditions, stabilizing them and consequently suppressing the transcription of GmWRKY100 to delay senescence. The genetic and biochemical experiments seemed to be done carefully, and the manuscript was written in a clear and concise manner. Some of the findings by the authors are novel and potentially significant, and advance our understanding on how a GmCRY1s-GmDELLAs-GmWRKY100 regulatory cascade regulates LBL-induced leaf senescence in soybean.

There are some suggestions that might improve this manuscript:

1. Xu et al. recently reported that AtCRY1 physically interacts with AtDELLA proteins in a blue light-dependent manner, leading to their dissociation from SLEEPY1 (SLY1) and inhibiting GA-induced degradation of DELLA proteins to control the photomorphogenesis of Arabidopsis (Xu et al., 2021). This paper should be cited and discussed in the manuscript.
2. In Yeast two-hybrid (Y2H) assays (Fig. 2b), the protein-protein interactions of SD/- LWHA (5 mM 3-AT) selection medium are pretty weak, and the Y2H results of SD/- LWHA or SD/- LWHA (with lower concentration 3-AT) should be exhibited. The interaction of full-length GmCRY1b with DELLA proteins should be included in Fig. 2b. The results of Fig. 2b should show the overall photo, the white lines between should be deleted. AtRGA full-length/Prey often exhibits severe self-activation in Y2H assays, and truncated AtRGA/Bait can solve this problem. Is there a self-activation problem in the Y2H assays (Fig. 2b) with GmRGAA/b full-length/Prey?
3. In Fig. 2d, soybean hairy root callus tissue was used in CoIP assays. In case this study focuses on soybean leaf senescence, it is better to use stable transgenic soybean leaves for CoIP assays.
4. Negative controls should be present in BiFC analysis and soybean protoplasts should be used instead of Arabidopsis protoplasts (Fig. 2e). This blue light-dependent interaction between GmCRY1b and DELLA proteins was abrogated under LBL conditions in BiFC assay in protoplasts of Arabidopsis (Fig. 2e), the protein levels of GmCRY1b-nYFP, GmRGAA-cCFP, and GmRGA-b-cCFP under LBL conditions should be detected to demonstrate normal expression.
5. The contents of active GA in leaves gradually decrease during leaf senescence. Exogenous application of GA delays leaf senescence, while exogenous application of PAC (GA inhibitors) promotes leaf senescence, suggesting that GA is a senescence-impeding hormone (Schipper et al., 2007; Yu et al., 2009; Jibrán et al., 2013). DELLA-repressed mutants like *gai-t6*, *rga-t2*, *rgl1-1*, *rgl2-1*, and *ga1-3 gai-t6 rga-t2 rgl1-1 rgl2-1* (abbreviated as Q-DELLA/*ga1-3*) showed an early-senescence phenotype compared with the wild type (WT), whereas the mutants (*ga1-3*) in which biosynthesis of GA was blocked and DELLA protein was highly accumulated were showing delayed senescence. SAG12 and SAG29 were upregulated in Q-DELLA/*ga1-3* and downregulated in *ga1-3* plants (Chen et al. 2014.). It has also been demonstrated that DELLA proteins RGL1 and RGA interact with TFs WRKY45 and WRKY6, respectively, suppressing leaf senescence (Chen et al., 2017b; Zhang et al., 2018a). The reported inconsistent functions of GA suggest that GA might indirectly regulate leaf senescence through crosstalks in an antagonistic or synergistic manner, which depends on the plant species or specific mutation backgrounds (Guo et al., 2021). The authors of this manuscript shows that the mRNA level of senescence-positive regulator gene GmWRKY100 was upregulated in *Gmrgas-dm* mutants compared to wild type (Fig. 5a), and 10 μ M GA3 treatment increased GmWRKY100 mRNA levels (Fig. 5b). These results suggesting that gibberellin promotes plant senescence in soybean. One experiment that the authors might want to add to bolster their manuscript is to detect the senescence phenotype of the wild type (WT) soybean leaves treated with GA3, gibberellin inhibitor (PAC), and measure the mRNA levels of SAG reporter genes such as GmSAG12, GmSAG13, GmSAG113. The author should also show the senescence phenotype of GmGA2ox7a overexpression soybean leaves.
6. In order to show the regulation relationship between GA and blue light on soybean leaf senescence, it would be helpful to present the leaf senescence phenotype of the *Gmrgas-dm* mutant upon LBL treatment and investigate whether LBL treatment can promote leaf senescence even in the absence of DELLA proteins. Meanwhile, the leaf senescence phenotype of the GmCRY1s *qm* mutant with GA3 treatment should be detected, and the mRNA levels of WRKY100 should be measured to investigate whether GA3-induced WRKY100 expression is dependent on GmCRY1s.

7. The dual-luciferase (LUC) reporter assay results demonstrate that GmCRY1b, GmRGAA, and GmRGAb effectively suppressed the relative LUC activity compared to the GUS control protein under white light (Fig. 5e). The author has previously demonstrated that GmCRY1b interacts with DELLA proteins GmRGAA and GmRGAb in response to blue light. So there is a possibility that GmCRY1b interacts with DELLA proteins to regulate the expression of WRKY100 jointly. It seems necessary to co-expression p35S: GmCRY1b, p35S: GmRGAA/b structures, and ProGmWRKY100: LUC structure in the dual-luciferase (LUC) reporter assay and test whether GmCRY1b and DELLA protein jointly regulate WRKY100 promoter activity. Moreover, in the transient dual-luciferase (LUC) reporter assay, soybean protoplasts should be used instead of Arabidopsis mesophyll protoplasts.
8. Gmwrky100 knockout soybean mutants show delayed leaf senescence and a higher yield performance (Fig. 6). For potential applications in agriculture, it would be necessary to compare the whole growth period of the Gmwrky100 mutants and wild-type under natural field conditions.
9. It would be helpful to present soybean yield trait characteristics of GmCRY1s-qm and GmCRY1b-YFP-5 (Supplementary Fig.5), Gmrgas-dm mutant, and GmRGAb-OE plants (Supplementary Fig.15) under natural field conditions.
10. The expression levels of GmCRY1b overexpression transgenic lines GmCRY1b YFP-1 and CRY1b YFP-5 should be presented.
11. Similar to Fig. 4c, relative transcript levels of GmCRY1s and GmRGAs in wild-type TL1 cultivar at the indicated leaf age under long-day conditions should be detected to understand their relationship better.
12. The author used Arabidopsis protoplasts in Fig. 2e and Fig. 5e, and the cultivation conditions, seedling age, and leaf type of Arabidopsis need to be specified in the manuscript.
13. Line 211-"Fig. 1a" should be "Supplementary Fig.13".
14. Line 273-"(Fig. 5e)" should be added at the end of the sentence.
15. The meaning of the percentage numbers in Fig. 1f and Fig.4h should be indicated.

Reviewer #2 (Remarks to the Author):

The manuscript by Li et al. shows that lowering the amount of blue light received by a soybean leaf (LBL) reduces its chlorophyll content while increasing the expression of senescence-associated genes (SAG). Furthermore, in a mutant line lacking the blue-light photoreceptor cryptochrome 1 (cry1) chlorophyll levels are constitutively low, and the expression of SAG is high (Fig. 1, S1-S7). Cry1 physically interacts with DELLA proteins RGAA and RGAb in Y2H experiments, and this interaction is blue-light dependent in co-IP experiments using soybean hairy root callus tissue, BiFC experiments in Arabidopsis protoplasts and split-luciferase complementation imaging assays in tobacco (Fig. 2, S9-S10). In Arabidopsis protoplasts, DELLA proteins RGAA and RGAb decrease their abundance in response to LBL and show elevated levels in the cry1 mutant (Fig. 2). The mutant deficient in RGAs showed reduced chlorophyll and increased SAG expression, overexpression of cry1 caused the opposite phenotype and the combination of cry1 overexpression in the RGA mutant background yielded intermediate levels of chlorophyll and reduced SAG expression (Fig. 3, S13, S15-S16). LBL increases the expression of the GmWRKY100 transcription factor gene and the mutant lacking GmWRKY100 shows greener leaves and reduced SAG expression, but the chlorophyll levels still respond to LBL (Fig. 4). The mutant deficient in RGAs shows elevated levels of expression of GmWRKY100 and RGAb associates with the promoter of the GmWRKY100 gene in ChIP assays using p35S:GmRGAb×Flag transgenic lines of soybean (Fig. 5). Based on these results the authors propose a model in which, under high blue light, cry1 stabilises RGAs, allowing their association with the promoter of the GmWRKY100 gene to repress its expression and consequently the expression of SAG and the progression of leaf senescence. Under LBL, the activity of cry1 decreases and the abundance of RGAs proteins decays, favouring the expression of the GmWRKY100, SAG expression and senescence.

The paper involves a significant number of carefully conducted experiments and the proposed model is interesting. However, I have some major concerns.

1. One of the key observations reported here is that cry1 interacts with RGA proteins. Similar

observations have been published for other species (PMCID: PMC8364249, PMID: 33971366, PMCID: PMC8577155) but I could not find these papers in the reference list.

2. The biological implications of the cry1-RGA interaction are not clearly elucidated. LBL reduces and cry1 enhances the abundance of RGAs, which is consistent with the model, but LBL-cry1 could affect RGA stability by other mechanisms not included in the proposed model. First, cry1 physically interacts with COP1 (PMCID: PMC5648270) a component of multimeric E3 ligases that physically interacts with RGA, targeting it to degradation in the proteasome (PMCID: PMC7306988). Second, cry could affect the levels of active gibberellins (GA, PMID: 36843173), and hence the abundance of RGA via the canonical GA pathway. Therefore, there are three pathways potentially linking cry1 to RGA: LBL-cry1-RGA, LBL-cry1-COP1-RGA, and LBL-cry1-....-GA-GID1-RGA. We do not know the actual contribution of the pathway proposed here (LBL-cry1-RGA).

3. All the experiments showing changes in RGA abundance were conducted in Arabidopsis protoplasts. Senescence is a process that can be strongly affected by the developmental context. According to the methods section the authors have a p35S:GmRGA \times Flag transgenic line of soybean that would be suitable to investigate at least the response to LBL in the plant organ context. However, this line was apparently not used for that purpose.

4. RGA associates with the GmWRKY100 promoter, but this does not mean that it directly inhibits its expression. In many cases DELLA proteins interact with transcription factors affecting their abundance, binding capacity, or intrinsic activity. RGA could act indirectly by regulating the activity of an unidentified transcription factor that binds the GmWRKY100 promoter.

5. The combination of cry1 overexpression and the RGA mutation yielded apparently additive effects on chlorophyll levels and reduced SAG expression (close to the expression levels observed in the cry1 overexpressor) (Fig. 3). The mutation causing RGA deficiency should eliminate the cry1 effect mediated by RGA, but this mutation did not substantially reduce the effect of cry1, suggesting that a large portion of the effect of cry1 does not require RGA. The mutant lacking GmWRKY100 shows greener leaves and reduced SAG expression, but the chlorophyll levels still respond to LBL as the wild type (Fig. 4h), suggesting that increased GmWRKY100 does not make a big contribution to the decay in chlorophyll levels. These results do not necessarily contradict the proposed model, but they certainly do not add experimental support for it.

6. The results of Fig. 5e are difficult to interpret based on the proposed model. The last combination (cry1, no RGAs) responds to blue light, suggesting that the protoplasts have enough endogenous RGAs to mediate the response. The second and third combinations (RGAs, no cry1) respond to blue light, suggesting that the protoplasts have enough endogenous cry1 to mediate the response. However, in the control condition (no cry1, no RGAs) there is no response to blue light suggesting that the endogenous levels of cry1 and/or RGAs are not sufficient for the response. It is not clear how these observations can be integrated.

7. If cry1 mediates the responses to LBL, the cry1 mutant should show reduced responses to LBL as observed for chlorophyll content (Fig. 1f). However, this has not been tested for the expression of SAG and for the abundance of RGA proteins.

8. According to Fig. 6, the GmWRKY100 mutants yield more than the wild type. It is not clear whether this results from their reduced senescence (which would be relevant to the other results of the paper) because these mutants also have a higher number of pods that could contribute to yield.

Point-by-point response to Reviewers:

REVIEWER' COMMENTS

Reviewer #1 (Remarks to the Author):

The manuscript by Li et al. reported the possible molecular mechanisms of GmCRY1s, The blue light receptors, in regulating low blue light-induced leaf senescence in soybean. They provided molecular and genetic evidence to demonstrate that GmCRY1s interact with DELLA proteins under light-activated conditions, stabilizing them and consequently suppressing the transcription of *GmWRKY100* to delay senescence. The genetic and biochemical experiments seemed to be done carefully, and the manuscript was written in a clear and concise manner. Some of the findings by the authors are novel and potentially significant, and advance our understanding on how a GmCRY1s-GmDELLAs-GmWRKY100 regulatory cascade regulates LBL-induced leaf senescence in soybean.

There are some suggestions that might improve this manuscript:

1. Xu et al. recently reported that AtCRY1 physically interacts with AtDELLA proteins in a blue light-dependent manner, leading to their dissociation from SLEEPY1 (SLY1) and inhibiting GA-induced degradation of DELLA proteins to control the photomorphogenesis of Arabidopsis (Xu et al.,2021). This paper should be cited and discussed in the manuscript.

Response: We are grateful for your reminder. We have included citations for this paper (Xu et al., 2021) and two other relevant studies (Zhong, et al., 2021; Yan, et al., 2021) and discussed their implications in our revised manuscript (Lines 355 - 357).

“Recently, the physical interactions between CRY1 and DELLA proteins have also been reported in Arabidopsis and wheat, suggesting that the CRY-DELLA signaling module is conserved among various species⁵¹⁻⁵³”.

2. In Yeast two-hybrid (Y2H) assays (Fig. 2b), the protein-protein interactions of SD/- LWHA (5 mM 3-AT) selection medium are pretty weak, and the Y2H results of SD/- LWHA or SD/- LWHA (with lower concentration 3-AT) should be exhibited. The interaction of full-length GmCRY1b with DELLA proteins should be included in Fig. 2b. The results of Fig. 2b should show the overall photo, the white lines between should be deleted. AtRGA full-length/Prey often exhibits severe self-activation in

Y2H assays, and truncated AtRGA/Bait can solve this problem. Is there a self-activation problem in the Y2H assays (Fig. 2b) with GmRGAA/b full-length/Prey?

Response: We appreciate your careful review and thank you for providing constructive suggestions. You are correct that there is also a self-activation problem in the Y2H assays with the full-length proteins of GmRGAb (**Supplementary Fig. 11, 12**). Following your suggestions, we have conducted Y2H assays to investigate the interaction of GmCRY1b and different truncated forms of GmCRY1b with DELLA proteins with lower concentration 3-AT (0 mM, 1 mM, 2 mM, 3 mM). The results showed that those positive interactions are relatively strong with 1 mM 3-AT (**Fig. 2b**, overall photo without white lines).

Given that the GmCRY1b full-length/Bait exhibits strong self-activation (**Supplementary Fig. 11**), it is not feasible to examine the interaction between the full length of GmCRY1b and DELLA proteins using the Y2H auxotrophic assay. Therefore, we used Y2H liquid assay, co-immunoprecipitation (Co-IP) assay and bimolecular fluorescence complementation (BiFC) assay to determine the interaction between full-length GmCRY1b and DELLA proteins (**Fig. 2c-e**).

3. In Fig. 2d, soybean fairy root callus tissue was used in CoIP assays. In case this study focuses on soybean leaf senescence, it is better to use stable transgenic soybean leaves for CoIP assays.

Response: Thank you for your valuable suggestions. We agree that using leaves of stable transgenic soybean expressing both GmCRY1b and DELLA proteins would be better than using fairy root callus tissue for Co-IP assays. However, obtaining such stable transgenic soybean is challenging. Therefore, we expressed the DELLA protein (GmRGAA or GmRGAb) in the *GmCRY1b-YFP-1* transgenic soybean using the soybean leaf injection method as described in PMID: 34591638. As shown in Fig. 2d, Flag-GmRGAA or Flag-GmRGAb successfully co-immunoprecipitated with GmCRY1b-YFP under blue light conditions, but not under dark conditions. The detailed procedure of soybean leaf injection was added in the Methods section (Line 496 - 504) as follows:

“The seedlings of GmCRY1b-YFP-1 transgenic line were grown under long-day conditions (16 h light/8 h dark) for seven days. The fully expanded unifoliate leaves were wiped with a brush⁴⁸. The Agrobacterium strain GV3101 transformed with 3×Flag-GmRGAA or 3×Flag-GmRGAb overexpression vectors were resuspended with infiltration buffer (10 mM MES PH 5.6, 200 μM acetosyringone) to OD600 = 1, and then pressure-infiltrated into the lower epidermis of the leaves using a vacuum pump until the leaves were completely wet. The transformed soybean seedlings were

recovered under continuous darkness for one day and then grown under normal long-day conditions.”

4. Negative controls should be present in BiFC analysis and soybean protoplasts should be used instead of Arabidopsis protoplasts (Fig. 2e). This blue light-dependent interaction between GmCRY1b and DELLA proteins was abrogated under LBL conditions in BiFC assay in protoplasts of Arabidopsis (Fig. 2e), the protein levels of GmCRY1b-cCFP, GmRGAA-nYFP, and GmRGAB-nYFP under LBL conditions should be detected to demonstrate normal expression.

Response: We appreciate your comments and suggestions. We have performed the BiFC experiments using soybean mesophyll protoplasts, and provided the negative controls in the revised manuscript (**Fig. 2e, Supplementary Fig. 13a**). Additionally, we performed immunoblot analysis to demonstrate the normal expression of GmCRY1b-cCFP, GmRGAA-nYFP, and GmRGAB-nYFP under WL and LBL conditions (**Supplementary Fig. 13b**).

5. The contents of active GA in leaves gradually decrease during leaf senescence. Exogenous application of GA delays leaf senescence, while exogenous application of PAC (GA inhibitors) promotes leaf senescence, suggesting that GA is a senescence-impeding hormone (Schippers et al., 2007; Yu et al., 2009; Jibrán et al., 2013). DELLA-repressed mutants like gai-t6, rga-t2 rgl1-1 rgl2-1, and ga1-3 gai-t6 rga-t2 rgl1-1 rgl2-1 (abbreviated as Q-DELLA/ga1-3) showed an early-senescence phenotype Compared with the wild type (WT), whereas the mutants (ga1-3) in which biosynthesis of GA was blocked and DELLA protein was highly accumulated were showing delayed senescence. SAG12 and SAG29 were upregulated in Q-DELLA/ga1-3 and downregulated in ga1-3 plants (Chen et al. 2014.) . It has also been demonstrated that DELLA proteins RGL1 and RGA interact with TFs WRKY45 and WRKY6, respectively, suppressing leaf senescence (Chen et al., 2017b; Zhang et al., 2018a). The reported inconsistent functions of GA suggest that GA might indirectly regulate leaf senescence through crosstalks in an antagonistic or synergistic manner, which depends on the plant species or specific mutation backgrounds (Guo et al., 2021). The authors of this manuscript shows that the mRNA level of senescence-positive regulator gene GmWRKY100 was upregulated in Gmrgas-dm mutants compared to wild type (Fig. 5a), and 10 μ M GA3 treatment increased GmWRKY100 mRNA levels (Fig. 5b). These results suggesting that gibberellin promotes plant senescence in soybean. One experiment that the authors might want to add to bolster their manuscript is to detect the senescence phenotype of the wild type (WT) soybean leaves treated with GA3, gibberellin inhibitor (PAC),

and measure the mRNA levels of SAG reporter genes such as GmSAG12, GmSAG13, GmSAG113. The author should also show the senescence phenotype of GmGA2ox7a overexpression soybean leaves.

Response: Thank you for your helpful suggestions. Gibberellins (GA) are an essential class of phytohormones that regulate a variety of plant growth and development processes. However, the role of GA in regulating leaf senescence process is still controversial as GA has been reported to regulate leaf senescence positively or negatively in different plant species, or in the same plant at different developmental stages, or in specific mutant backgrounds. Therefore, determining the role of GA in regulating leaf senescence in soybean is of great importance for this study.

Following your suggestions, we treated wild-type plants with 10 μ M GA₃ and 5 μ M PAC, respectively. Phenotypic analysis demonstrated that exogenous GA₃ treatment significantly accelerated the process of leaf senescence, whereas PAC (a gibberellin inhibitor) treatment delayed this process (**Supplementary Fig. 16a-d**). Additionally, GA₃ treatment induced the expression of senescence-associated genes, including *GmSAG12*, *GmSAG13*, and *GmSAG113*, but PAC treatment repressed their expression (**Supplementary Fig. 16e**). Furthermore, overexpression of *GmGA2ox7a* resulted in an obviously delayed leaf senescence phenotype compared to wild-type plants (**Supplementary. Fig 17**). Taken together, these results suggest that GA functions as a senescence-promoting phytohormone in soybean.

6. In order to show the regulation relationship between GA and blue light on soybean leaf senescence, it would be helpful to present the leaf senescence phenotype of the Gmrgas-dm mutant upon LBL treatment and investigate whether LBL treatment can promote leaf senescence even in the absence of DELLA proteins. Meanwhile, the leaf senescence phenotype of the GmCRY1s qm mutant with GA3 treatment should be detected, and the mRNA levels of WRKY100 should be measured to investigate whether GA3-induced WRKY100 expression is dependent on GmCRY1s.

Response: Thank you very much for this valuable comment. Our results showed that LBL-induced leaf senescence was less pronounced in *Gmrgas-dm* mutant compared to wild-type plants (**Supplementary Fig. 26**), supporting the idea that DELLA proteins inhibit the LBL-induced leaf senescence process.

Following your suggestion, we also investigated whether GmCRY1s are involved in GA-mediated leaf senescence. We treated the *GmCRY1s-qm* mutant, *GmCRY1b-YFP-5* overexpression line, and wild type plants with 10 μ M GA₃ exogenously. Phenotype analysis showed that *GmCRY1b* overexpression could clearly repress GA₃-induced leaf senescence, whereas *GmCRY1s-qm* mutant exhibited enhanced GA₃-induced leaf senescence phenotype (**Supplementary Fig. 18**).

We further examined the GA₃-induced *GmWRKY100* expression in *Gmcry1s-qm* mutant, and found that although GA₃ could induce *GmWRKY100* expression in *Gmcry1s-qm* mutant, the extent was much less than in wild type plants (**Supplementary Fig. 31**). These results suggested that GA₃-induced *WRKY100* expression is at least partially dependent on GmCRY1s.

7. The dual-luciferase (LUC) reporter assay results demonstrate that GmCRY1b, GmRGAA, and GmRGAb effectively suppressed the relative LUC activity compared to the GUS control protein under white light (Fig. 5e). The author has previously demonstrated that GmCRY1b interacts with DELLA proteins GmRGAA and GmRGAb in response to blue light. So there is a possibility that GmCRY1b interacts with DELLA proteins to regulate the expression of WRKY100 jointly. It seems necessary to co-expression p35S: GmCRY1b, p35S: GmRGAA/b structures, and ProGmWRKY100: LUC structure in the dual-luciferase (LUC) reporter assay and test whether GmCRY1b and DELLA protein jointly regulate WRKY100 promoter activity. Moreover, in the transient dual-luciferase (LUC) reporter assay, soybean protoplasts should be used instead of Arabidopsis mesophyll protoplasts.

Response: Thank you for your valuable comment and helpful suggestions. Following your advice, we performed the dual-luciferase (LUC) reporter assay in soybean protoplasts and found that LBL could induce the transcriptional activity of *GmWRKY100* promoter in soybean.

Furthermore, we co-expressed GmCRY1b and the DELLA proteins (GmRGAA or GmRGAb) in soybean mesophyll protoplasts, respectively, and found that their co-expression repress the transcription activity of *GmWRKY100* promoter to a similar extent as the expression of only DELLA proteins (**Fig. 5e**). we speculate that the hyperaccumulation of DELLA protein is sufficient to repress the transcription activity of *GmWRKY100* promoter. Therefore, the co-expression GmCRY1b and DELLA protein did not significantly repress the transcription activity of *GmWRKY100* promoter compared to the expression of only DELLA proteins.

8. Gmwrky100 knockout soybean mutants show delayed leaf senescence and a higher yield performance (Fig. 6) . For potential applications in agriculture, it would be necessary to compare the whole growth period of the Gmwrky100 mutants and wild-type under natural field conditions.

Response: Thank you for your valuable comments. Based on your suggestion, we planted the *Gmwrky100* mutant and wild-type plants under natural field conditions. We did not observe any significant difference in the growth rate between the *Gmwrky100* knockout mutant and wild-type plants (**Supplementary Fig. 33**). However, the process

of leaf senescence in *Gmwrky100* mutant lagged significantly behind the wild-type plants (**Supplementary Fig. 29**).

9. It would be helpful to present soybean yield trait characteristics of *Gmcry1s-qm* and *GmCRY1b-YFP-5* (Supplementary Fig.5), *Gmrgas-dm* mutant, and *GmRGAb-OE* plants (Supplementary Fig.15) under natural field conditions.

Response: Thank you very much for your valuable comments. We have described the yield trait characteristics of the *Gmcry1s-qm* mutant and *GmCRY1b-YFP-5* overexpression line under natural field conditions in our previous publication (PMID:33249237). Overexpression of *GmCRY1b* improved soybean performance under high-density planting conditions, such as a much lower lodging percentage, and an increased number of efficient branches. Statistical analysis showed that the yield per plant of the overexpression lines increased by nearly 50% compared with that of wild-type control. In this revised manuscript, we have examined the yield trait characteristics of the *Gmrgas-dm* mutant and *GmRGAb* overexpression line under natural field conditions. We found that the *Gmrgas-dm* mutant and the *GmRGAb* overexpression line exhibit a significantly higher and shorter plant height, respectively, than wild-type plants. However, there was no significant change in node number, branch number, and total grains weight per plant between the *Gmrgas-dm* mutant and the *GmRGAb* overexpression line and wild-type plants under natural field conditions (**Supplementary Fig. 25**).

10. The expression levels of *GmCRY1b* overexpression transgenic lines *GmCRY1b YFP-1* and *CRY1b YFP-5* should be presented.

Response: Thank you very much for your kind reminder. We have examined the mRNA and protein level of *GmCRY1b* in the *GmCRY1b-YFP-1* and *CRY1b-YFP-5* transgenic lines. All these results demonstrate the respective overexpression levels of *GmCRY1b-YFP* in these lines (**Supplementary Fig. 4**).

11. Similar to Fig. 4c, relative transcript levels of *GmCRY1s* and *GmRGAs* in wild-type *TL1* cultivar at the indicated leaf age under long-day conditions should be detected to understand their relationship better.

Response: We appreciate your constructive comments regarding the improvement of our manuscript. We investigated the transcript levels of *GmCRY1b*, *GmRGAA* and *GmRGAb* at the indicated leaf age under long-day conditions as shown in Fig. 4c. Interestingly, similar to the expression pattern of *GmWRKY100*, the transcript levels of *GmCRY1b*, *GmRGAA* and *GmRGAb* also significantly increased when leaf senescence was induced (**Supplementary Fig. 28**). We added more sentences to the revised

manuscript (Lines 270-275) to clarify these findings:

“The transcriptional level of GmWRKY100 was observed to increase in association with the onset of leaf senescence (Fig. 4c), suggesting that GmWRKY100 promotes leaf senescence in soybean. Notably, the expression levels of GmCRY1b, GmRGAA and GmRGAB also increased as leaf senescence progressed (Supplementary Fig. 28), suggesting the existence of a negative feedback regulation among GmCRY1s, DELLAs, and GmWRKY100 in the process of leaf senescence.”

12. The author used Arabidopsis protoplasts in Fig. 2e and Fig. 5e, and the cultivation conditions, seedling age, and leaf type of Arabidopsis need to be specified in the manuscript.

Response: We appreciate your careful review. Following your suggestion, we used soybean protoplasts instead of Arabidopsis protoplasts in the BIFC assays to determine the interaction between GmCRY1b and DELLA proteins. We added the detailed information to the revised manuscript (Lines 556-559), as follows:

“Soybean seedlings were grown under short-day (8 h light/16 h dark) conditions, at a light intensity of 120-180 $\mu\text{mol m}^{-2} \text{s}^{-1}$ and a temperature of 26 °C. Mesophyll protoplasts were isolated from the unifoliolate leaves of soybean and transformed following the reported procedure.”

13. Line 211-“Fig. 1a” should be “Supplementary Fig.13”.

Response: Revised

14. Line 273-“(Fig. 5e)” should be added at the end of the sentence.

Response: Revised

15. The meaning of the percentage numbers in Fig. 1f and Fig.4h should be indicated.

Response: Thank you for your reminder regarding this issue. We have added the meaning of the percentage numbers to the revised manuscript (Lines 788-789, 852-854), as follows:

“The percentage decrease in chlorophyll content under WL compared to LBL is indicated by the values above the respective p values”.

Reviewer #2 (Remarks to the Author):

The manuscript by Li et al. shows that lowering the amount of blue light received by

a soybean leaf (LBL) reduces its chlorophyll content while increasing the expression of senescence-associated genes (SAG). Furthermore, in a mutant line lacking the blue-light photoreceptor cryptochrome 1 (cry1) chlorophyll levels are constitutively low, and the expression of SAG is high (Fig. 1, S1-S7). Cry1 physically interacts with DELLA proteins RGAa and RGAb in Y2H experiments, and this interaction is blue-light dependent in co-IP experiments using soybean hairy root callus tissue, BiFC experiments in Arabidopsis protoplasts and split-luciferase complementation imaging assays in tobacco (Fig. 2, S9-S10). In Arabidopsis protoplasts, DELLA proteins RGAa and RGAb decrease their abundance in response to LBL and show elevated levels in the cry1 mutant (Fig. 2). The mutant deficient in RGAs showed reduced chlorophyll and increased SAG expression, overexpression of cry1 caused the opposite phenotype and the combination of cry1 overexpression in the RGA mutant background yielded intermediate levels of chlorophyll and reduced SAG expression (Fig. 3, S13, S15-S16). LBL increases the expression of the GmWRKY100 transcription factor gene and the mutant lacking GmWRKY100 shows greener leaves and reduced SAG expression, but the chlorophyll levels still respond to LBL (Fig. 4). The mutant deficient in RGAs shows elevated levels of expression of GmWRKY100 and RGAb associates with the promoter of the GmWRKY100 gene in ChIP assays using p35S:GmRGAb×Flag transgenic lines of soybean (Fig. 5). Based on these results the authors propose a model in which, under high blue light, cry1 stabilises RGAs, allowing their association with the promoter of the GmWRKY100 gene to repress its expression and consequently the expression of SAG and the progression of leaf senescence. Under LBL, the activity of cry1 decreases and the abundance of RGAs proteins decays, favouring the expression of the GmWRKY100, SAG expression and senescence.

The paper involves a significant number of carefully conducted experiments and the proposed model is interesting. However, I have some major concerns.

1. One of the key observations reported here is that cry1 interacts with RGA proteins. Similar observations have been published for other species (PMCID: PMC8364249, PMID: 33971366, PMCID: PMC8577155) but I could not find these papers in the reference list.

Response: Thank you for your kind reminder. We sincerely apologize for not including these relevant publications for citation in our previous version of the manuscript. We have added these reference papers to the revised manuscript (Lines 355-357, Reference number: 51, 52, 53) as follows:

“Recently, the physical interactions between CRY1 and DELLA proteins have also been

reported in Arabidopsis and wheat, suggesting that the CRY-DELLA signaling module is conserved among various species⁵¹⁻⁵³.”

2. The biological implications of the cry1-RGA interaction are not clearly elucidated. LBL reduces and cry1 enhances the abundance of RGAs, which is consistent with the model, but LBL-cry1 could affect RGA stability by other mechanisms not included in the proposed model. First, cry1 physically interacts with COPI (PMCID: PMC5648270) a component of multimeric E3 ligases that physically interacts with RGA, targeting it to degradation in the proteasome (PMCID: PMC7306988). Second, cry could affect the levels of active gibberellins (GA, PMID: 36843173), and hence the abundance of RGA via the canonical GA pathway. Therefore, there are three pathways potentially linking cry1 to RGA: LBL-cry1-RGA, LBL-cry1-COPI-RGA, and LBL-cry-....-GA-GIDI-RGA. We do not know the actual contribution of the pathway proposed here (LBL-cry1-RGA).

Response: We appreciate your constructive comments and apologize for not clearly elucidating the biological implications of the cry1-RGA interaction. We have added more discussion in the revised manuscript (Lines 354-370) as follows:

“DELLA proteins are central components in the control of plant growth responses to adapt to environmental changes. Recently, physical interactions between CRY1 and DELLA proteins have been reported in Arabidopsis and wheat, suggesting that the CRY-DELLA signaling module is conserved among various species⁵¹⁻⁵³. Here, we found that GmCRY1b also interacts with GmRGAs (GmRGAa and GmRGA b) in a blue light-dependent manner in soybean (Fig. 2d), and loss-of-function of GmCRY1s largely abolishes the accumulation of GmRGAs (Fig. 2i). Furthermore, LBL reduces the activity of GmCRY1s and the protein abundance of GmRGAs, suggesting that the blue light-dependent interaction between GmCRY1s and GmRGAs is crucial for the protein stability of GmRGAs possibly through two mechanisms. Firstly, GmCRY1s may protect GmRGAs by interfering with COPI, a component of multimeric E3 ligases that physically interacts with RGA (PMCID: PMC5648270), targeting it to degradation in the proteasome (PMCID: PMC7306988). Second, GmCRY1s could affect the levels of active gibberellins (PMID: 36843173), and hence the abundance of RGA via the canonical GA pathway. Given that LBL treatments only induce local leaf senescence rather than systematic leaf senescence (Fig. 1a-c), we hypothesis that the LBL-GmCRY1-RGA pathway, rather than LBL-induced up-regulation of GA levels, may play a major role in the induction of local senescence of leaves that are shaded by the upper vegetative canopy.”

3. All the experiments showing changes in RGA abundance were conducted in

Arabidopsis protoplasts. Senescence is a process that can be strongly affected by the developmental context. According to the methods section the authors have a p35S:3 ×Flag-GmRGAbtransgenic line of soybean that would be suitable to investigate at least the response to LBL in the plant organ context. However, this line was apparently not used for that purpose.

Response: Thank you for your important and valuable comments. Following your suggestion, we conducted immunoblot experiments to test how RGA protein responds to LBL conditions in the stable transgenic line *Flag-GmRGAb-1*. Similar to the trend of DELLA proteins GmRGAA and GmRGAB in soybean hairy root callus, immunoblot analysis showed that the protein levels of GmRGAB in the stable transgenic line *Flag-GmRGAb -1* decrease significantly in response to LBL treatment (**Supplementary Fig. 19**). The results were added in the revised manuscript (Lines 205-207) as follows:

“Consistent with the results observed in soybean hairy root callus, GmRGAb protein also showed a similar reduction in response to LBL in the Flag-GmRGAb-1 stable transgenic line (Supplementary Fig. 19)”

4. RGA associates with the GmWRKY100 promoter, but this does not mean that it directly inhibits its expression. In many cases DELLA proteins interact with transcription factors affecting their abundance, binding capacity, or intrinsic activity. RGA could act indirectly by regulating the activity of an unidentified transcription factor that binds the GmWRKY100 promoter.

Response: Thank you so much for your valuable comments. We strongly agree with your opinion that RGA might regulate the transcript levels of *GmWRKY100* in an indirect manner. DELLA proteins, lacking a canonical DNA-binding domain, function as transcriptional repressor via direct protein-protein interactions with multiple transcription factors belonging to various hormonal and environmental signaling pathways, such as PHYTOCHROME INTERACTING FACTORS (PIFs), JASMONATE ZIM-domain (JAZ) and INDE-TERMINATE DOMAIN (IDD) family proteins, and further repress their transcriptional activation activity. Our future work will focus on identifying the transcription factor that directly bind to the *GmWRKY100* promoter and co-regulate the *GmWRKY100* transcript level with DELLA proteins direct interaction.

5. The combination of cry1 overexpression and the RGA mutation yielded apparently additive effects on chlorophyll levels and reduced SAG expression (close to the expression levels observed in the cry1 overexpressor) (Fig. 3). The mutation causing RGA deficiency should eliminate the cry1 effect mediated by RGA, but this mutation did not substantially reduce the effect of cry1, suggesting that a large portion of the

effect of cry1 does not require RGA. The mutant lacking GmWRKY100 shows greener leaves and reduced SAG expression, but the chlorophyll levels still respond to LBL as the wild type (Fig. 4h), suggesting that increased GmWRKY100 does not make a big contribution to the decay in chlorophyll levels. These results do not necessarily contradict the proposed model, but they certainly do not add experimental support for it.

Response: Thank you for providing valuable comments. We also realized that the RGA mutation only partially rescued the delayed leaf senescence in the GmCRY1b overexpression background. This is possibly because other DELLA proteins are functionally redundant with GmRGAA and GmRGAB. The soybean genome encodes four DELLA proteins: GmRGAA (*Glyma.05140400*), GmRGAB (*Glyma.08G095800*), GmRGLA (*Glyma.11G216500*), and GmRGLB (*Glyma.18G040000*). The amino acid sequence similarity of each pair of DELLA protein gene is as high as 89%-92%. Each single gene mutation of DELLA proteins shows no significant change in the process of leaf senescence compared with wild-type plants in soybean, whereas the double mutant of *GmRGAA* and *GmRGAB* exhibits a clearly premature leaf senescence phenotype (**Supplementary Fig. 22**). These results suggested that DELLA proteins function redundantly in regulating leaf senescence and other two DELLA proteins GmRGLA and GmRGLB might also participate in the regulation of CRY1-mediated leaf senescence.

The LBL-induced decrease in chlorophyll content is 24.4% and 20.7% in *Gmwrky100-1* and *Gmwrky100-7*, respectively, which is significantly lower than that in wild-type plants at 28.6%, suggesting that GmWRKY100 at least partially accounts for LBL-induced leaf senescence.

6. The results of Fig. 5e are difficult to interpret based on the proposed model. The last combination (cry1, no RGAs) responds to blue light, suggesting that the protoplasts have enough endogenous RGAs to mediate the response. The second and third combinations (RGAs, no cry1) respond to blue light, suggesting that the protoplasts have enough endogenous cry1 to mediate the response. However, in the control condition (no cry1, no RGAs) there is no response to blue light suggesting that the endogenous levels of cry1 and/or RGAs are not sufficient for the response. It is not clear how these observations can be integrated.

Response: Thank you for pointing out this issue. We agree that it is unclear that why the *GmWRKY100* promoter shows no response to LBL in Arabidopsis protoplasts in the control condition (no cry1, no RGA). One possibility is that overexpression of cry1 or RGA is necessary for the response of *GmWRKY100* promoter to LBL in Arabidopsis protoplasts. To address this issue, we performed the dual-luciferase (LUC) reporter assay in soybean protoplasts. The results show that the transcription activity of

GmWRKY100 promoter could be induced under LBL conditions in soybean mesophyll protoplasts even in the control condition (no cry1, no RGA) (Fig. 5e).

7. If *cry1* mediates the responses to LBL, the *cry1* mutant should show reduced responses to LBL as observed for chlorophyll content (Fig. 1f). However, this has not been tested for the expression of SAG and for the abundance of RGA proteins.

Response: Thank you so much for pointing out this issue. We have tested the transcript levels of SAGs in the *Gmcry1s-qm* mutant, *Gmcry2s-tm* mutant and wild-type plants under white light and LBL conditions. The results demonstrated that the response of SAGs to LBL was significantly reduced in the *Gmcry1s-qm* mutant compared with wild-type plants and the *Gmcry2s-tm* mutant (Supplementary Fig. 8).

8. According to Fig. 6, the *GmWRKY100* mutants yield more than the wild type. It is not clear whether this results from their reduced senescence (which would be relevant to the other results of the paper) because these mutants also have a higher number of pods that could contribute to yield.

Response: We appreciate your constructive comment and thank you very much for your effort to improve our manuscript. The *GmWRKY100* mutants *Gmwrky100-1* and *Gmwrky100-7* exhibit clearly delayed leaf senescence phenotype under greenhouse and natural field conditions (Fig. 4d, Supplementary Fig. 29). Furthermore, the *Gmwrky100* mutants showed an increase in grain yield. An examination of the agronomic traits related to grain yield in *Gmwrky100-1* and *Gmwrky100-7* showed that, except for the increase in pods number per plant (Fig. 6), there were no significant difference in other agronomic traits including flowering time, plant height, node number and branch number during the whole growth period between wild-type plants and *Gmwrky100* mutants (Fig. 6, and Supplementary Fig. 22, 33). While it is difficult to determine whether *GmWRKY100* directly or indirectly affects pod number, it is reasonable to deduce that the absence of *GmWRKY100* extends the functional photosynthesis period of soybean leaves, thereby contributing to the production of more pods due to adequate energy supply.

Reviewer #1 (Remarks to the Author):

The authors carefully considered my comments and made appropriate changes. I only found one issue that authors may want to address:

According to Fig. 5e of the revised manuscript, the transcription activity of GmWRKY100 promoter could be induced under LBL conditions in soybean mesophyll protoplasts even in the control condition (only GUS, no CRY1, no RGA), but not in Arabidopsis mesophyll protoplasts in the same control condition (Fig. 5e of the first version: 416746_0_art_file_7395612_rq22tz). Why? The necessary explanations should be presented.

Reviewer #2 (Remarks to the Author):

The authors have considered all my previous concerns. In some cases, their responses are satisfactory but in others further action is required.

Previous comment 1: The response is satisfactory.

Previous comment 2. The authors are presenting as a single mechanism two of the alternative mechanisms that I had suggested in my previous comment. Although not tested here, it is a reasonable possibility. However, I do not think that the hypothesis that blue light affects senescence via changes in gibberellin can be disregarded because the senescence response is local. Gibberellin could change locally or simply require local factors to induce senescence. Unless the authors can show that low blue light does not increase the levels of gibberellins in the leaves, they should simply acknowledge that with current data it is impossible to discriminate between the two possibilities.

Regarding the same issue, please move the position of the reference as indicated: "Firstly, GmCRY1s may protect GmRGAs by interfering with COP1 (PMCID: PMC5648270), a component of multimeric E3 ligases that physically interacts with RGA (PMCID: PMC5648270), targeting it to degradation in..."

Previous comment 3. The response is satisfactory. However, in a paper dealing with the role of DELLA proteins in leaf senescence, the kinetics of DELLA proteins in the leaves should be within the main figures and the confirmatory data in calluses should be among the supplementary figures and not vice versa (as in the current version of the paper).

Previous comment 4. The authors agree that the effect of DELLA is likely to be indirect. However, this is not acknowledged in the text. Actually, in the model of Figure 5h DELLA directly binds DNA. The likely involvement of one or more transcription factors not identified here should be acknowledged in the text and in the schematic representation of the model to avoid misinterpretations.

Previous comment 5. The authors argue that the DELLA mutation has little effect in the cry1 overexpressor background because cry1 could still affect the accumulation of chlorophyll by affecting the remaining DELLAs. I agree that this is a reasonable explanation. However, it is still necessary to provide support to the model where DELLAs act downstream cry1. Perhaps it would be possible to repeat the chlorophyll measurements in the presence of added gibberellin to sensitise the system to the mutation of DELLAs in the presence of the cry1 overexpressor. Furthermore, the authors argue that the effects of blue light in the wrky100 mutants is larger than in the wild type because in the mutants the impact is 24.4% and 20.7%, whereas in the wild type it is 28.6%. This is possible, but adequate statistical treatment would require the demonstration that there is significant interaction between genotypes and light treatments in a two-way ANOVA.

Previous comment 6. The response is satisfactory. However, do not just show that all the conditions respond to low blue light because that is not very informative. The figure should highlight that DELLAs have a statistically significant effect in the presence of blue light and not under low blue light. Acknowledge that there would be enough cry1 in the protoplasts and for that

reason transient expression makes no difference.

Previous comment 7. The response is satisfactory. Perhaps the result should be within the main body of figures.

Previous comment 8. The problem with the proposed explanation is that by the time strong leaf senescence is initiated in soybean crops the number of pods is already fixed. If leaf senescence had been the factor limiting grain yield, I would have expected a larger impact on the weight of 1000 grains rather than on the number of pods. The yield results are likely unrelated to the mechanisms described here. I fully understand that ending this work with a field experiment showing the agronomic impact of the mechanism described in the laboratory experiments is very nice but there is no certainty that that is the case and this could be misleading for the reader not familiar with the crop.

REVIEWER COMMENTS

Reviewer #1 (Remarks to the Author):

The authors carefully considered my comments and made appropriate changes. I only found one issue that authors may want to address:

According to Fig. 5e of the revised manuscript, the transcription activity of GmWRKY100 promoter could be induced under LBL conditions in soybean mesophyll protoplasts even in the control condition (only GUS, no CRY1, no RGA), but not in Arabidopsis mesophyll protoplasts in the same control condition (Fig. 5e of the first version: 416746_0_art_file_7395612_rq22tz). Why? The necessary explanations should be presented.

Response: Thank you for bringing this issue to our attention. We acknowledge that the *GmWRKY100* promoter is induced by LBL in soybean mesophyll protoplasts but not in Arabidopsis protoplasts under control conditions (only GUS, no CRY1, no RGA). Therefore, we chose soybean protoplasts for the experiment. We hypothesize that the endogenous levels of RGA or CRY1 in Arabidopsis protoplasts may be insufficient to elicit an LBL-induced response from the *GmWRKY100* promoter. Our data support this hypothesis, as the extent of the LBL-induced response of the *GmWRKY100* promoter is positively correlated with RGA abundance in both Arabidopsis (**Fig. 5e** in the first version) and soybean protoplasts (**Fig. 5e** in this version).

Reviewer #2 (Remarks to the Author):

The authors have considered all my previous concerns. In some cases, their responses are satisfactory but in others further action is required.

Previous comment 1: The response is satisfactory.

Response: Thanks for your careful review.

Previous comment 2. The authors are presenting as a single mechanism two of the alternative mechanisms that I had suggested in my previous comment. Although not tested here, it is a reasonable possibility. However, I do not think that the hypothesis that blue light affects senescence via changes in gibberellin can be disregarded because the senescence response is local. Gibberellin could change locally or simply require local factors to induce senescence. Unless the authors can show that low blue light does not increase the levels of gibberellins in the leaves, they should simply

acknowledge that with current data it is impossible to discriminate between the two possibilities. Regarding the same issue, please move the position of the reference as indicated: “Firstly, GmCRY1s may protect GmRGAs by interfering with COP1 (PMCID: PMC5648270), a component of multimeric E3 ligases that physically interacts with RGA (PMCID: PMC5648270), targeting it to degradation in...”

Response: Thank you for your important and valuable comments. We have carefully considered your suggestion and revised the discussion (Lines 373-378) as follows:

“We speculate that GmCRY1s may protect GmRGAs through two potential mechanisms. Firstly, GmCRY1s may protect GmRGAs by interfering with COP1(PMCID: PMC5648270), a component of multimeric E3 ligases that physically interacts with RGA(PMCID: PMC5648270), targeting it to degradation in the proteasome (PMCID: PMC7306988) . Second, GmCRY1s may affect the levels of active gibberellins (PMID:36843173) , and hence the abundance of RGA via the canonical GA pathway. Future experiments will be necessary to discriminate between the two possibilities.”

Previous comment 3. The response is satisfactory. However, in a paper dealing with the role of DELLA proteins in leaf senescence, the kinetics of DELLA proteins in the leaves should be within the main figures and the confirmatory data in calluses should be among the supplementary figures and not vice versa (as in the current version of the paper).

Response: Thank you for your valuable suggestions. We have incorporated your feedback and added the kinetics of DELLA proteins in the leaves to the main figures (**Fig. 2f, g**), and the kinetics of DELLA proteins in calluses have been placed in the supplementary figures (**Supplementary Fig. 18**).

Previous comment 4. The authors agree that the effect of DELLA is likely to be indirect. However, this is not acknowledged in the text. Actually, in the model of Figure 5h DELLA directly binds DNA. The likely involvement of one or more transcription factors not identified here should be acknowledged in the text and in the schematic representation of the model to avoid misinterpretations.

Response: Thank you for your reminder regarding this issue. We have carefully considered your suggestion and addressed the indirect role of DELLA in binding DNA by adding an unknown “X” protein in the schematic model representation (Fig. 5h) and providing further explanation in the revised manuscript (Lines 319-321) as follows:

“Since DELLA proteins lack a canonical DNA-binding domain, it is likely that unidentified transcriptional factors play a role in regulating the activity of the GmWRKY100 promoter, while DELLA proteins act as repressors by interacting with these factors.”

Previous comment 5. The authors argue that the DELLA mutation has little effect in the cry1 overexpressor background because cry1 could still affect the accumulation of chlorophyll by affecting the remaining DELLAs. I agree that this is a reasonable explanation. However, it is still necessary to provide support to the model where DELLAs act downstream cry1. Perhaps it would be possible to repeat the chlorophyll measurements in the presence of added gibberellin to sensitise the system to the mutation of DELLAs in the presence of the cry1 overexpressor. Furthermore, the authors argue that the effects of blue light in the wrky100 mutants is larger than in the wild type because in the mutants the impact is 24.4% and 20.7%, whereas in the wild type it is 28.6%. This is possible, but adequate statistical treatment would require the demonstration that there is significant interaction between genotypes and light treatments in a two-way ANOVA.

Response: We appreciate your careful review and thank you for providing constructive suggestions. Based on your feedback, we exogenously treated the *GmCRY1b* overexpressor with gibberellin and performed phenotypic analysis, which showed that gibberellin treatment could largely repress the effect of *GmCRY1b* overexpression in delaying leaf senescence (**Supplementary Fig. 26**). These results suggest that GmCRY1s repress leaf senescence largely through the action of DELLA proteins in soybean. We have incorporated these results into the revised manuscript (Lines 253-257) as follows:

“Further application of 10 μM GA₃ accelerated the process of leaf senescence in GmCRY1b-YFP-1 and Gmrgas-dm/GmCRY1b-YFP-1 plants to a similar extent to that as observed in wild-type plants (Supplementary Fig. 26). This implies that other DELLA proteins may function alongside GmRGAa and GmRGAb, which are genetically downstream of GmCRY1s, to regulate leaf senescence in soybean.”

As you suggested, we utilized a two-way analysis of variance (two-ANOVA) to investigate the role of the *GmWRKY100* gene in LBL-induced leaf senescence. Since age-induced leaf senescence in the later stage of leaf development could obscure the effect of LBL-induced leaf senescence, we measured the chlorophyll content of wild-type plants and *Gmwrky100* mutants at different time points over a week following nine days of LBL treatment (**Supplementary Fig. 32**). Our analysis revealed that the interaction between genotypes and light treatment was significant at least on days 12 and 13 of LBL treatment (**Response Table 1**). Therefore, these findings support that the *GmWRKY100* gene plays a significant role in LBL-induced leaf senescence. We have included these results in the revised manuscript (Lines 286-293):

“To determine the function of GmWRKY100 in LBL-induced leaf senescence, we

grew wild-type plants and Gmwrky100 mutants under WL and LBL conditions, respectively. Phenotypic analysis revealed a reduced LBL-induced leaf senescence in the Gmwrky100 mutant compared to the wild type (Fig. 4g, Supplementary Fig. 32). The statistical analysis of two-way ANOVA indicated a significant interaction between genotypes and light treatments in terms of chlorophyll content and senescence marker gene GmSAG13 (Fig. 4h, i). These results suggest that the GmWRKY100 gene plays significant role in LBL-induced leaf senescence.”

Response Table 1

Days after LBL-treatment	P value of interaction
9	0.9141
10	0.7587
11	0.6932
12	0.0152
13	0.0252
14	0.1657
15	0.0603

Previous comment 6. The response is satisfactory. However, do not just show that all the conditions respond to low blue light because that is not very informative. The figure should highlight that DELLAs have a statistically significant effect in the presence of blue light and not under low blue light. Acknowledge that there would be enough cry1 in the protoplasts and for that reason transient expression makes no difference.

Response: Thank you very much for this valuable comment. We have modified the figure (**Fig. 5e**) and incorporated your suggestions into the revised manuscript (Lines 321-323):

“To be noted, adding GmCRY1b made no observable impact on the transient expression assays (Fig. 5e), likely indicating a sufficient presence of CRY1 in the protoplasts.”

Previous comment 7. The response is satisfactory. Perhaps the result should be within the main body of figures.

Response: Thank you for your helpful suggestions. We have incorporated your feedback and placed the expression data of senescence-associated mark genes with the main body of figures (**Fig. 1g**).

Previous comment 8. The problem with the proposed explanation is that by the time

strong leaf senescence is initiated in soybean crops the number of pods is already fixed. If leaf senescence had been the factor limiting grain yield, I would have expected a larger impact on the weight of 1000 grains rather than on the number of pods. The yield results are likely unrelated to the mechanisms described here. I fully understand that ending this work with a field experiment showing the agronomic impact of the mechanism described in the laboratory experiments is very nice but there is no certainty that that is the case and this could be misleading for the reader not familiar with the crop.

Response: We appreciate the reviewer's helpful comments. Our results indicate that soybean leaf senescence is not a uniform process, and that unifoliate leaves begin to senesce three weeks after sowing under field conditions. Floral transition occurs seven weeks after sowing. We also observed no significant differences in weight of 100 grains between wild-type plants and the *Gmwrky100* mutant, indicating that the increase in grain weight per plant is mainly due to an increase in pod number (**Fig. 6g**). Unlike rice and wheat, the position of soybean pods is not fixed, and pod numbers are a yield-determining trait that is more susceptible to internal cues and the external environment (PMID:32171732). Our study suggests that increasing the functional photosynthetic period has the potential to increase soybean pod numbers by providing sufficient energy to nourish more pods in the early stage of pod development. We hope that this clarifies the implications of our work for readers who may not be familiar with soybean crops.

Reviewer #2 (Remarks to the Author):

The authors have satisfactorily addressed all my concerns.